# Discovery and characterization of genes conferring natural resistance to the antituberculosis antibiotic capreomycin

Shu-Ing Toh [1,2], Johan Elaine Keisha[1], Yung-Lin Wang[3], Yi-Chi Pan[1], Yu-Heng Jhu[1], Po-Yun Hsiao[1], Wen-Ting Liao[1,2], Po-Yuan Chen [1,4], Tai-Ming Ko[1,4] & Chin-Yuan Chang [1,5,6 ✉]

Metagenomic-based studies have predicted an extraordinary number of potential antibiotic-resistance genes (ARGs). These ARGs are hidden in various environmental bacteria and may become a latent crisis for antibiotic therapy via horizontal gene transfer. In this study, we focus on a resistance gene *cph*, which encodes a phosphotransferase (Cph) that confers resistance to the antituberculosis drug capreomycin (CMN). Sequence Similarity Network (SSN) analysis classified 353 Cph homologues into five major clusters, where the proteins in cluster I were found in a broad range of actinobacteria. We examine the function and antibiotics targeted by three putative resistance proteins in cluster I via biochemical and protein structural analysis. Our findings reveal that these three proteins in cluster I confer resistance to CMN, highlighting an important aspect of CMN resistance within this gene family. This study contributes towards understanding the sequence-structure-function relationships of the phosphorylation resistance genes that confer resistance to CMN.

[1] Department of Biological Science and Technology, National Yang Ming Chiao Tung University, Hsinchu 30010 Taiwan, ROC. [2] Institute of Molecular Medicine and Bioengineering, National Yang Ming Chiao Tung University, Hsinchu 30010 Taiwan, ROC. [3] Genomics Research Center, Academia Sinica, Taipei 11529 Taiwan, ROC. [4] Institute of Bioinformatics and Systems Biology, National Yang Ming Chiao Tung University, Hsinchu 30010 Taiwan, ROC. [5] Center for Intelligent Drug Systems and Smart Bio-devices, National Yang Ming Chiao Tung University, Hsinchu 30010 Taiwan, ROC. [6] Department of Biomedical Science and Environment Biology, Kaohsiung Medical University, Hsinchu 80708 Taiwan, ROC. ✉email: cycytl@nycu.edu.tw

Many antibiotics have been discovered and subsequently utilized in clinical therapy or animal husbandry. However, anthropogenic activities, overuse/misuse of antibiotics, and spontaneous evolution via gene mutation have developed pathogens with antibiotic resistance and continue to be a threat to human health[1,2]. Antibiotic-resistance genes (ARGs) are widespread in nature and can be disseminated by one or more distinct gene transfer mechanisms[3,4]. Notably, ARGs are able to be transmitted by horizontal gene transfer from nonpathogens to pathogens. This mechanism for ARG dissemination has been a major pathway in the evolution of pathogens against antibiotics[3,4]. Fortunately, recent understanding of various antibiotic-resistance mechanisms provided insights into design and development of antibiotics to cope with the drug-resistance problems[5–8].

*Mycobacteria tuberculosis* is an archetypical human pathogen; it evolved with the human race and as much as one-third of world population is estimated to be currently infected[9,10]. Over 225 million cases of tuberculosis (TB) and 79 million TB-related deaths are estimated from 1998 to 2030[11]. Unfortunately, this pathogen has evolved into the multidrug-resistant tuberculosis (MDR-TB) and even the extremely drug-resistant tuberculosis (XDR-TB) forms, which are insusceptible to four or more of the front-line drugs[12]. Capreomycin (CMN) and viomycin (VIO) are nonribosomal peptide (NRP) natural products (Fig. 1) discovered and isolated from *Saccharothrix mutabilis* subsp. *capreolus* (previously named *Streptomyces capreolus*) and *Streptomyces* sp. ATCC11861, respectively[13–16]. CMN and VIO are known to bind to the 23S rRNA and 16S rRNA at the ribosome interface, consequently blocking the formation of the translation initiation complex and preventing the translocation of tRNA[17–20]. Both drugs belong to the tuberactinomycin family of antibiotics and CMN, in particular, has historically been a pivotal second-line drug for MDR-TB treatment. However, the growing concerns regarding treatment failures and relapses compared to regimens without it have diminished its preference. As a result, CMN is no longer recommended for MDR-TB treatment regimens. The World Health Organization (WHO) delisted CMN from the essential drugs roster in 2019[21]. CMN now stands as a drug for conditional use.

Several genes are correlated with CMN and VIO resistance. The CMN- and VIO-producing strains possess intrinsic self-resistance determinants within or outside their biosynthetic gene clusters (BGCs) that protect themselves from their toxic products. Self-resistance determinants act as vital reservoirs and could influence the evolution of pathogens toward drug resistance[22–24]. In the CMN-producing strain, *S. capreolus*, three genes, *cph*, *cac*, and *cmnU*, were reported to be involved in self-resistance mechanisms: (i) *cph*, found within the CMN BGC, encodes a phosphotransferase that inactivates CMN by phosphorylation of

the hydroxyl group of Ser[16,25–27], (ii) *cac*, located outside the CMN BGC, encodes an *N*-acetyltransferase that inactivates CMN by acetylation of the β-Lys moiety[25,26], and (iii) *cmnU*, also found within the CMN BGC, encodes a putative methyltransferase that was proposed to methylate 16S rRNA, which becomes insensitive to CMN via antibiotic target modification[16]. In contrast to the CMN BGC, only *vph*, a *cph* homologous gene, was found within the VIO BGC and reported to be involved in self-resistance mechanisms in the VIO-producing strain[15,16].

The BGC of CMN and its biosynthesis, both the formation of the cyclic NRP backbone and tailing modifications, have been extensively studied[28–31]. Our previous biochemical study and structural determination of Cph supported that it provides two molecular mechanisms of self-resistance: (i) Ser phosphorylation of CMN IIA and (ii) sequestration of CMN IIB, an Ala derivative of CMN IIA (Fig. 1), in the substrate-binding pocket[27]. Cph alone endows *S. capreolus* with CMN resistance by both chemical modification and physical sequestration. Intriguingly, homologs of *cph* are broadly distributed in many bacteria species in nature. This raises an interesting question: do the natural *cph* homologs confer resistance to CMN? Furthermore, do these *cph* homologs show substrate specificity to CMN?

In this study, we aimed to understand whether the widespread *cph* homologs function as resistance genes and confer resistance to the CMN family of antibiotics. To achieve this, we first generated a SSN that classified the Cph family into five major clusters. Sequence analysis initially suggested that the proteins in one of the SSN clusters were CMN-resistance proteins. Biochemical and structural investigation indicated that the proteins in this SSN cluster confer resistance to CMN. This study systematically discovers and identifies the specific genes for antibiotic resistance in genetic databases and contributes towards understanding the sequence-structure-function relationships of the CMN-resistance proteins.

## Results and discussion

**Bioinformatics analysis revealing 130 Cph homologs for CMN resistance.** BLASTP analysis identified over 3000 Cph putative homologs in the NCBI GenBank non-redundant protein sequence database, with amino acid sequence identities ranging from 15% to 100%. Most putative Cph homologs identified through BLASTP analysis are annotated as phosphotransferases, including viomycin phosphotransferase, aminoglycoside phosphotransferase (APH), and macrolide phosphotransferase (MPH). However, the authentic function and targeted antibiotics of these homologs are not known. To identify phosphotransferases conferring resistance to the CMN family of antibiotics, we constructed an SSN using 353 homologs from the UniProt database, which had amino acid sequence identities

**Fig. 1 Structures of two congeners of capreomycin (CMN IIA and CMN IIB) and viomycin (VIO).** Ser residue for Cph phosphorylation and β-Lys residue for Cac acetylation in CMN IIA and CMN IIB are colored red and green, respectively.

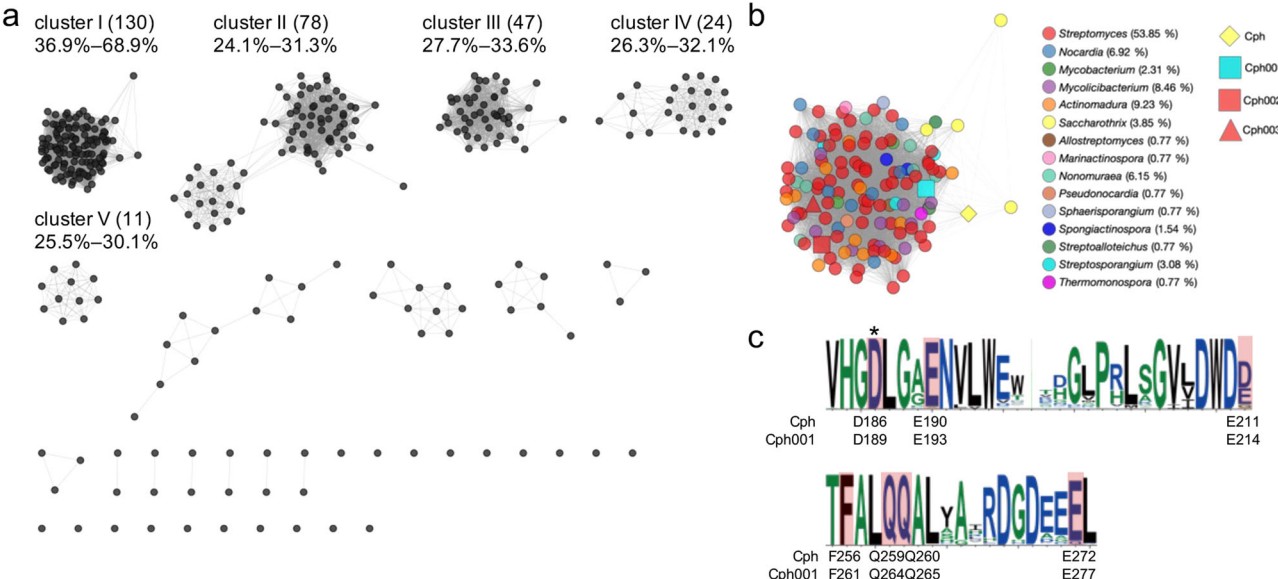

**Fig. 2 Bioinformatics analysis of the Cph homologs. a** The 353 Cph homologs are classified into five major clusters in an SSN at an *e* value of $10^{-55}$. The numbers enclosed in parentheses denote the protein quantity, while the values underneath signify the sequence identities with Cph. **b** The cluster I consists of 130 Cph homologs, which are found in a broad range of bacteria highlighted with different colors. **c** Sequence logos reveal the conserved residues for CMN binding among the Cph members in cluster I. Residues invovled in CMN binding are highlighted with a red background. The catalytic residue is marked with an asterisk (*) above the sequence logos. Below the sequence logos, residues associated with CMN binding and catalysis are displayed.

ranging between 20% and 70% with Cph. These Cph homologs were classified into five major clusters in the SSN at an *e* value threshold of $10^{-55}$ (Fig. 2a). Cph from the CMN-producing strain is located in a large cluster, cluster I, which consists of 130 Cph homologs with 36.9% to 68.9% sequence identities to Cph (Fig. 2b; Supplementary Table 1). Proteins found in the four other major clusters, II–V, share up to 33.6% sequence identities to Cph. The sequence alignments of Cph with proteins from each cluster can be found in Supplementary Figs. 1–12.

We previously solved the crystal structures of Cph and its complex with CMN or ATP[30]. Cph consists of two domains: an N-terminal lobe domain responsible for ATP binding and a C-terminal domain divided into a core subdomain and an α-helical subdomain for CMN binding. In the complex structure of Cph with CMN, CMN is bound by the side chains of Asp186, Glu190, and Glu211 at the core subdomain and Gln259, Gln260, and Glu272 at the α-helical subdomain through hydrogen bonding. Additionally, Phe256 interacts with the amide group of CMN through π-electron interactions[27]. Sequence analysis of proteins in cluster I revealed conservation of the CMN-binding residues (Fig. 2c; Supplementary Fig. 1); proteins in clusters II–V do not conserve these residues (Supplementary Figs. 2–12). This implicates that the Cph homologs in cluster I contain the putative CMN-binding site and may confer resistance to CMN. Intriguingly, the Cph homologs in cluster I were found in a broad range of actinobacteria, including *Mycobacterium*, *Mycolicibacterium*, and *Pseudonocardia* (Fig. 2b).

**Cph homologs in cluster I confer resistance to CMN.** To confirm whether proteins in cluster I also confer resistance to CMN, three proteins, Cph001 from *Streptosporangium roseum* (accession number: WP_012891209), Cph002 from *Streptomyces* sp. SID8364 (MYT78782), and Cph003 from *Streptomyces filamentosus* NRRL15998 (EWS93654), were selected for disk diffusion assays. These three bacterial genomes were sequenced; notably, none of them possess the CMN or VIO BGCs. Cph001, Cph002, and Cph003 share 49.02%, 48.16%, and 45.17% amino acid sequence identities to Cph, respectively (Supplementary

Fig. 13). The genes encoding these three proteins were synthesized and cloned into pET22b and used to transform *E. coli* BL21 Rosetta (DE3). Disk diffusion assay revealed that each of the *E. coli* strains harboring *cph*, *cph001*, *cph002*, or *cph003*, becomes resistant to CMN IIA at different levels (Fig. 3a).

**Activity assay and mutational analysis of Cph001 for CMN phosphorylation.** Cph is an *O*-phosphotransferase that transfers the γ-phosphate group of ATP or GTP to the hydroxyl group of CMN IIA. To further confirm their mechanism of resistance, we attempted to produce Cph001, Cph002, and Cph003 in *E. coli* and to purify to homogeneity for enzyme assays. However, Cph002 and Cph003 were not stable, undergoing aggregation and degradation during protein purification. Therefore, only Cph001 could be used for further biochemical characterization and structural determination (Supplementary Fig. 14). The Cph001 reaction was performed using a lactate dehydrogenase-coupled enzyme assay following the procedure previously used for Cph[27]. A decrease in absorbance at 340 nm was recorded for both Cph and Cph001 reactions in the presence of CMN IIA and ATP, indicating phosphorylation by Cph001 (Fig. 3b). We then performed mutagenesis on the codons coding for the conserved catalytic residue Asp189 to generate the Cph001[D189A] and Cph001[D189N] variants. Activity assays revealed that both mutations resulted in complete loss of phosphotransferase activity supporting that Asp189 in Cph001 plays a critical role for catalysis (Fig. 3b). The apparent Michaelis constant ($K_m$) of CMN IIA and turnover number ($k_{cat}$) of Cph001 with ATP as a phosphate donor were determined to be $0.88 \pm 0.20$ mM and $0.63 \pm 0.05$ min$^{-1}$, respectively, under pseudo-first-order conditions (Fig. 3c, i). Furthermore, Cph001 also accepted GTP as the phosphate donor; the apparent $K_m$ of CMN IIA and $k_{cat}$ of Cph001 in the presence of saturating GTP were determined to be $1.40 \pm 0.24$ mM and $0.36 \pm 0.03$ min$^{-1}$, respectively (Fig. 3c, ii). On the other hand, the apparent $K_m$ and $k_{cat}$ of ATP and GTP for Cph001 in the presence of saturating CMN IIA were determined to be $0.07 \pm 0.02$ mM and $0.58 \pm 0.03$ min$^{-1}$ and $0.18 \pm 0.03$ mM and $0.91 \pm 0.04$ min$^{-1}$, respectively (Fig. 3c, iii; Fig. 3c, iv). The $K_m$

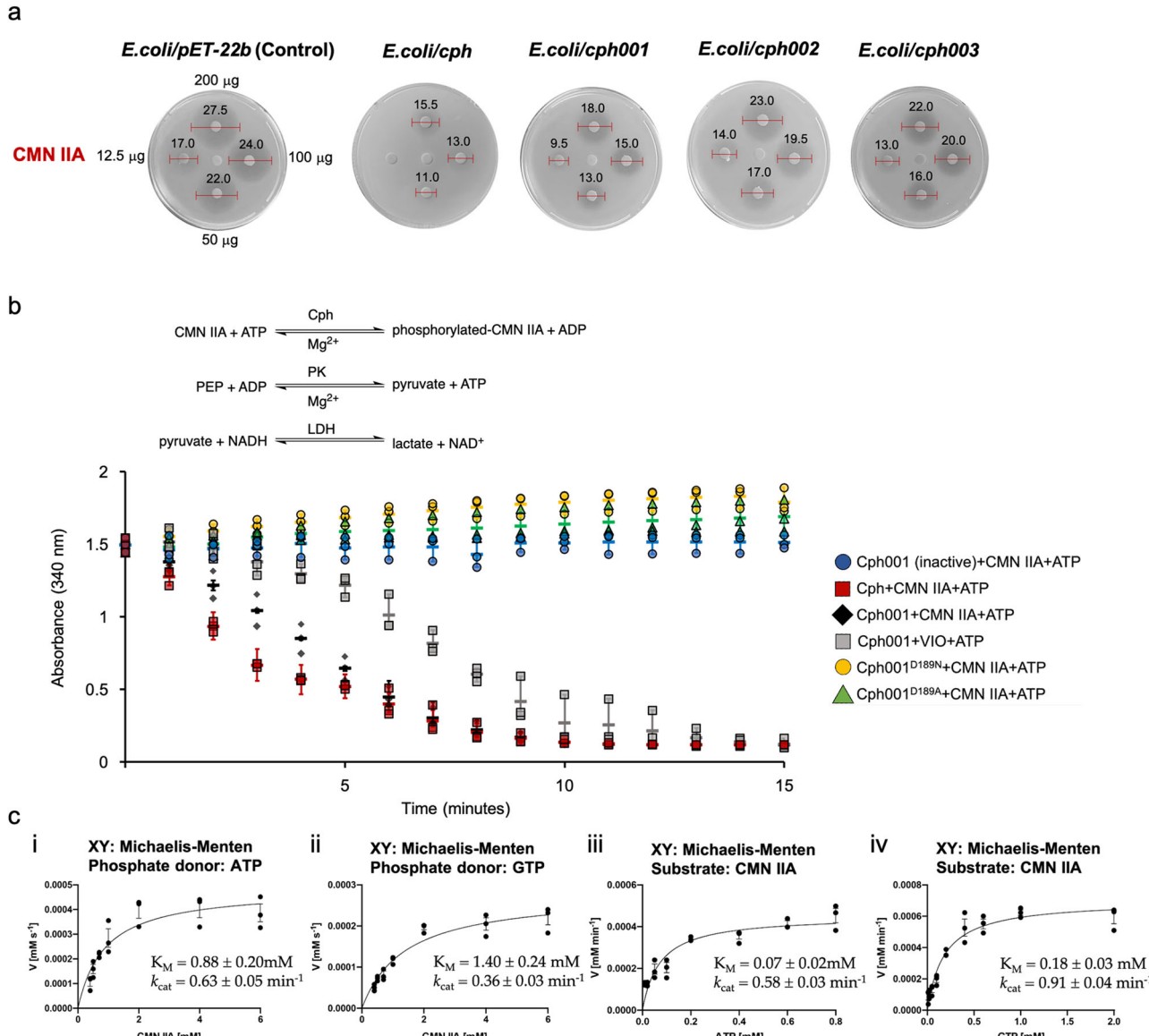

**Fig. 3 Disk diffusion assays and enzyme activity assays of Cph001. a** Disk diffusion assays of *E. coli* with or without *cph/cph001/cph002/cph003* gene against CMN IIA. The amounts of the antibiotics are loaded with the same concentration (center: 0 μg, left: 12.5 μg, down: 50 μg, right: 100 μg, and up: 200 μg) to each set of the disks. The numerical units indicated on the red bar used for measuring inhibition zones are in mm. **b** Scheme showing the steps of the couple enzyme assay used for Cph001 in this study. The reaction using the inactive-Cph001 (boiled enzyme) is negative control. The assays were performed by monitoring the absorbance at 340 nm due to the consumption of NADH to NAD⁺. PEP phosphoenolpyruvate, PK pyruvate kinase, LDH lactate dehydrogenase. **c** Michaelis-Menten kinetics of Cph001. The Cph001 reactions against series CMN IIA concentration with **i**. ATP and **ii**. GTP as a phosphate donor, and the Cph001 reactions against series **iii**. ATP and **iv**. GTP as a phosphate donor with CMN IIA. Each of the activity assays and kinetics was performed in triplicate ($n = 3$) and the data were presented as mean value ± standard error of the mean (SEM). The kinetics data were fit to a Michaelis-Menten model and the figures were generated by the software Prism 10.

for ATP was >2.5-fold tighter than the $K_m$ for GTP, indicating that Cph001 prefers ATP as a phosphate donor.

Previous kinetic studies have shown that the $K_m$ value of Cph for CMN IIA with ATP as a phosphate donor ($K_m = 0.46$ mM) is on the same level as that of Cph001, being less than twice the $K_m$ of Cph001. The environments of the CMN IIA binding sites in Cph and Cph001 are identical (the crystal structure of Cph001 will be discussed further below), leading to $K_m$ values that are on the same level. However, the $k_{cat}$ value of Cph for CMN IIA with ATP as a phosphate donor ($k_{cat} = 17.40$ min⁻¹) is about 27 times higher than that of Cph001. The more efficient catalysis of Cph phosphorylation stems from *cph* being the original self-resistance gene for CMN.

Unsurprisingly, Cph001 is incapable of phosphorylating CMN IIB due to the lack of the hydroxyl group as the phosphate acceptor (Fig. 1). However, similar to *cph*, *cph001* is capable of endowing *E. coli* with resistance to CMN IIB (Fig. 4a). We performed isothermal titration calorimetry (ITC) analysis to measure the dissociation constants ($K_D$) of CMNs. CMN IIA and CMN IIB bind to Cph001 with $K_D$ values 1.24 ± 0.08 μM and 55.20 ± 4.40 μM, respectively (Supplementary Fig. 15), supporting the binding of CMN IIB by Cph001 despite lacking the hydroxyl group. This result also echoed the observation of the disk diffusion assay that Cph001 confers resistance to CMN.

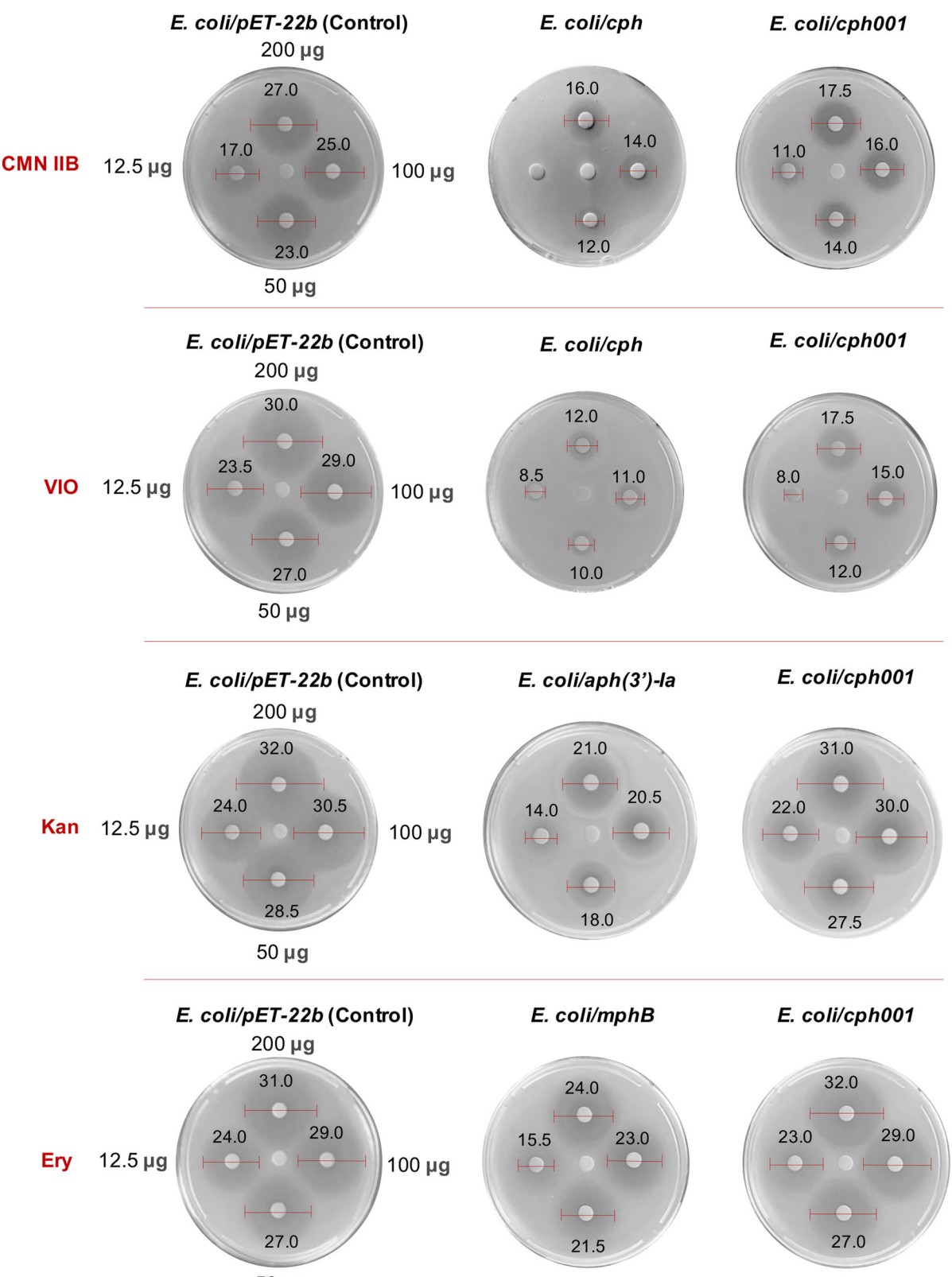

**Fig. 4 Disk diffusion assays *cph001* gene against the four antibiotics, CMN IIB, VIO, Kan, and Ery.** The amounts of the antibiotics are loaded with the same concentration (center: 0 µg, left: 12.5 µg, down: 50 µg, right: 100 µg, and up: 200 µg) to each set of the disks. The numerical units indicated on the red bar used for measuring inhibition zones are in mm. The gene *aph(3')-Ia* and *mphB* are known as conferring resistance to Kan and Ery, respectively[32,34].

**Cph001 confers cross-resistance to CMN and VIO**. Antibiotic resistance mediated by phosphotransferases was observed in many aminoglycoside and macrolide antibiotics[32–34]. All APHs, MPHs, Cph, and Cph001 belong to the phosphotransferase family. To investigate whether Cph001 confers cross-resistance to other classes of antibiotics or related antibiotics, an aminoglycoside antibiotic kanamycin (Kan), a macrolide antibiotic erythromycin (Ery), and VIO were subjected to the disk diffusion assay and couple enzyme assay. Based on the disk diffusion assay, *E. coli* harboring *cph001* are resistant to VIO but not to Kan and Ery (Fig. 4). We further performed the couple enzyme assay to evaluate the phosphorylation reaction of Cph001 to VIO, Kan, and Ery using ATP as a phosphate donor. Consistently, Cph001 is only capable of phosphorylating VIO (Fig. 3b), and thus Cph001 confers cross-resistance to the CMN analogous antibiotic VIO.

**Structural solution and refinement of Cph001 in apo and complex forms**. To understand the molecular details of Cph001 catalytic mechanism and CMN- and ATP/GTP-binding sites, we performed crystallization to determine the three-dimensional structure of Cph001. Degradation bands were observed on the SDS-PAGE for the wild-type Cph001 after a week of storage at 4 °C. This might be the reason why the wild-type Cph001 could not be crystallized. After multiple attempts to obtain the crystals of the wild-type Cph001, we switched to focus on the Cph001$^{D189N}$ variant, which is more stable than the wild-type and can be crystallized at 4 °C. Crystals of Cph001$^{D189N}$ were successfully obtained in a week and the diffraction data were collected at National Synchrotron Radiation Research Center (NSRRC) on beamlines 15A, 05A, or 07A. The molecular packing of Cph001$^{D189N}$ belongs to the orthorhombic space group $P2_12_12_1$. The crystal structure of Cph001$^{D189N}$ in apo form was determined by the molecular replacement (MR) method, with the structure of Cph (PDB entry 7F0A)[27] as the initial search model. Two polypeptide chains were found and built in the asymmetric unit (Fig. 5a). The final model of Cph001$^{D189N}$ was refined to a resolution of 2.3 Å. Furthermore, we obtained ligand complex structures by soaking CMN IIA, CMN IIB, VIO, ATP, or GTP into the crystals of Cph001$^{D189N}$. Seven structures of Cph001$^{D189N}$ complexes were determined by the MR method using the apo protein as the search model: Cph001$^{D189N}$-CMN IIA, Cph001$^{D189N}$-CMN IIB, Cph001$^{D189N}$-VIO, Cph001$^{D189N}$-ATP, Cph001$^{D189N}$-GTP, Cph001$^{D189N}$-CMN IIA/ATP, and Cph001$^{D189N}$-VIO/ATP. The final models of the binary complex structures, Cph001$^{D189N}$-CMN IIA, Cph001$^{D189N}$-CMN IIB, Cph001$^{D189N}$-VIO, Cph001$^{D189N}$-ATP, and Cph001$^{D189N}$-GTP, were refined to resolutions between 1.95 Å to 2.26 Å, and the ternary structures, Cph001$^{D189N}$-CMN IIA/ATP and Cph001$^{D189N}$-VIO/ATP, were refined to resolutions between 2.00 Å and 3.00 Å, respectively. The extra electron density in the putative active site can be identified and unambiguously fitted with the molecular conformation of CMN IIA, CMN IIB, VIO, ATP, and GTP in the corresponding complex structures (Supplementary Fig. 16). The summary of crystallographic data and refinement statistics of Cph001$^{D189N}$ and the seven complex structures are shown in Table 1.

**Crystal structures of Cph001 reveal molecular details for the binding sites of CMN/VIO and ATP/GTP**. The overall structures of Cph001 and Cph revealed an identical fold with a root-mean-square deviations (rmsd) of 0.97 Å over a superposition of 215 Cα atoms. The polypeptide chain of Cph001 is folded into two domains, an N-terminal lobe domain (Met1–Gly97) and a C-terminal domain (Ala98–Arg286). The C-terminal domain further divided into a core subdomain (Ala98–Ala140 and

Thr182–Ser246) and an α-helical subdomain (Pro141–Ile181 and Asp247–Arg286) (Fig. 5b). The complex structures, Cph001$^{D189N}$-CMN IIA, Cph001$^{D189N}$-CMN IIB, Cph001$^{D189N}$-ATP, and Cph001$^{D189N}$-GTP, revealed that CMN IIA and CMN IIB bind to the C-terminal domain while ATP and GTP bind to the N-terminal lobe domain opposite to CMN (Fig. 5b). Except for the β-lysine moiety that is flexible in the binding pocket, the pentapeptide backbones of CMN IIA and CMN IIB are found in a similar conformation. The residues involved in CMN binding are conserved between Cph001 and Cph (Supplementary Fig. 13). The side chains of Glu193 and Glu214 at the core subdomain and Gln264, Gln265, and Glu277 at the α-helical subdomain interact with CMN IIA or CMN IIB through hydrogen bonds (Fig. 5c). CMN IIA, compared to CMN IIB, contains an additional hydrogen bond between its hydroxyl group and Asp189 (mutated to Asn in the present structure) (Fig. 1; Fig. 5c). In addition, the side chain of Phe261 revealed a close contact with the amide group of CMN IIA or IIB through π–electron interactions (Fig. 5c). For the binding sites of ATP and GTP, the residues involved in ATP and GTP binding are highly conserved between Cph001 and Cph (Supplementary Fig. 13). The crystal structures of Cph001$^{D189N}$-ATP and Cph001$^{D189N}$-GTP revealed that ATP and GTP are bound in the same position, where the main-chain amides of Ser93 and Val95 interact with the base of ATP and GTP through hydrogen bonds (Fig. 5d). In a comparison of the ATP-binding environments of Cph001 and Cph, the side chain imidazole group of His31 associates with the β-phosphate group of ATP in Cph001 instead of the side chain guanidinium group of Arg41 to the α-phosphate group in Cph (Fig. 5f; Supplementary Fig. 13)[27]. Furthermore, the main chain of Gln29–His31 in a β-hairpin structure interacts with the β-phosphate group of ATP or GTP through hydrogen bonds in Cph001; however, this β-hairpin structure is far away from ATP in the complex structure of Cph with ATP (Fig. 5d; Fig. 5f).

$Mg^{2+}$ is widely recognized for its role in stabilizing the transition state of the phosphate group during phosphorylation[35]. The crystal structure of aminoglycoside-3′-phosphotransferase-IIIa [APH(3′)-IIIa, PDB entry 1L8T], an APH linked to resistance against Kan and neomycin, reveals that Asp208 and Asn195 serve as ligands coordinating with two $Mg^{2+}$ ions (Fig. 5g)[36]. The important role of Asp208 in stabilizing the transition state for catalytic reaction has been demonstrated[37]. Cph001 and APH(3′)-IIIa share a similar three-dimensional structure with an rmsd of 2.89 Å for Cα atom superposition. The $Mg^{2+}$-binding residues, Asp208 and Asn195 in APH(3′)-IIIa, correspond to Asp211 and Asn194 in Cph001, respectively, located near the triphosphate group of ATP (Fig. 5g). Furthermore, these two residues are conserved across almost all the proteins in the SSN analysis (Supplementary Fig. 1–12), suggesting their potential role in $Mg^{2+}$ binding.

Based on the disk diffusion assay and the couple enzyme assay mentioned previously, Cph001 confers cross-resistance to CMN and VIO. We then solved the crystal structure of Cph001$^{D189N}$-VIO. The complex structures Cph001$^{D189N}$-CMN IIA, Cph001$^{D189N}$-CMN IIB, and Cph001$^{D189N}$-VIO revealed that CMN and VIO occupy the same position with a similar conformation; the pentapeptide backbones are superimposed well while the β-lysine moiety is located on the opposite position between VIO and CMN (Fig. 5c; Fig. 5e). The β-lysine moiety of VIO extends into the α-helical subdomain while that of CMN extends towards the N-terminal lobe domain. Moreover, the additional hydroxyl group at the capreomycidine moiety of VIO points towards the protein surface without any specific interaction. The complex structure Cph001$^{D189N}$-VIO supports cross-resistance of Cph001 and provides molecular details of VIO-binding environment. ITC analysis revealed that VIO binds to

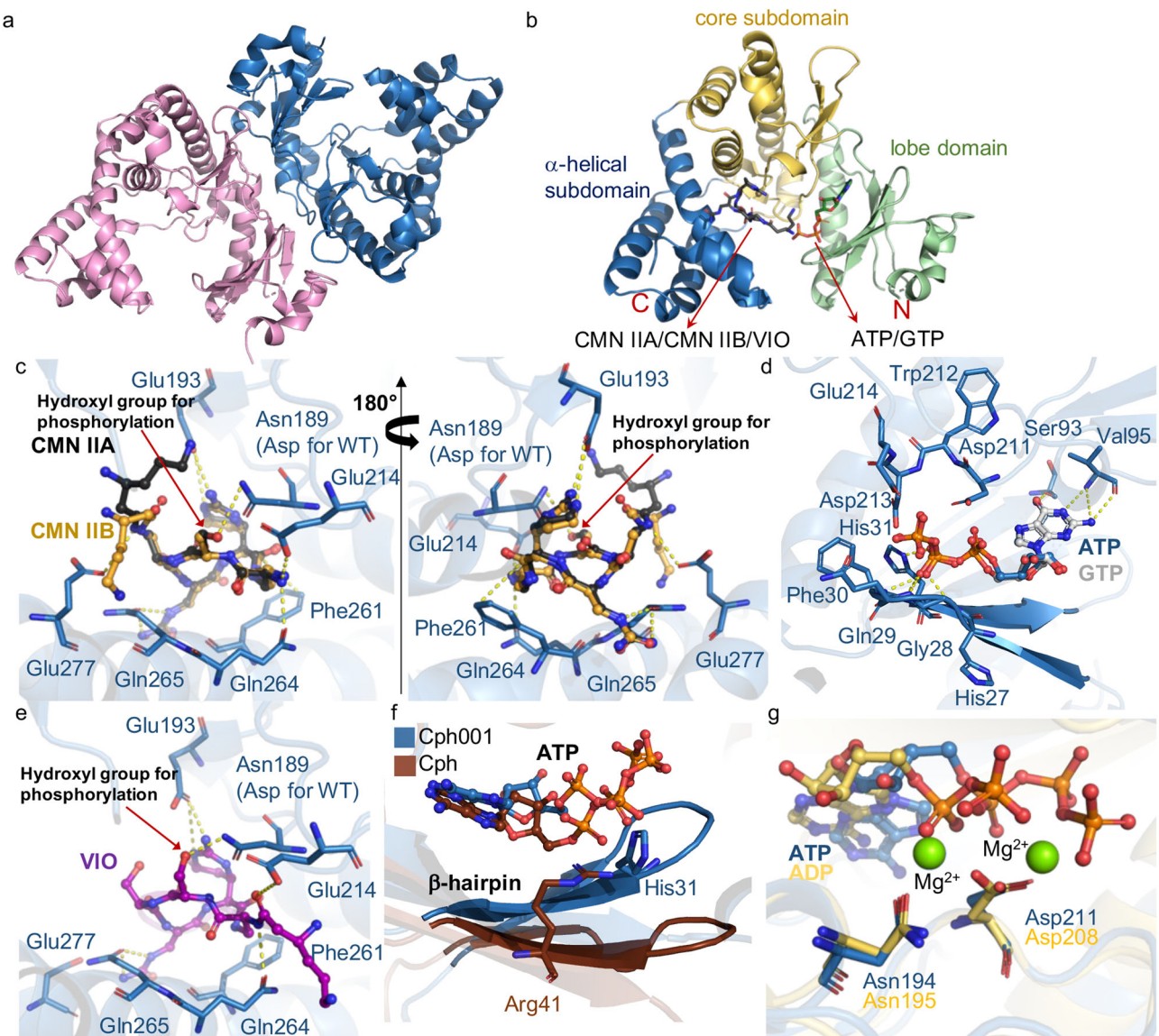

**Fig. 5 Crystal structures of Cph001. a** Dimer of Cph001. The two polypeptide chains are colored pink and blue. **b** Overall structure of one Cph001 subunit. The structure of Cph001 is divided into a lobe domain, a core subdomain, and an α-helical subdomain, which are colored green, yellow, and blue, respectively. **c** Local view of CMN IIA- and CMN IIB-binding site. CMN IIA and CMN IIB are depicted as balls and sticks in black and yellow. **d** Local view of ATP- and GTP-binding site. ATP and GTP are depicted as balls and sticks in blue and gray. **e** Local view of VIO-binding site. VIO is depicted as balls and sticks in purple. **f** Superposition of the ATP-binding site in Cph001 with that in Cph (PDB entry 7F0B). The β-hairpins of Cph001 and Cph are highlighted and colored blue and brown. **g** Superposition of the Mg$^{2+}$-binding site in APH(3′)-IIIa with that in Cph001. The crystal structures of APH(3′)-IIIa in complex with ADP and Mg$^{2+}$ (PDB entry 1L8T)[36] and Cph001$^{D189N}$-ATP are colored yellow and blue, respectively. The two Mg$^{2+}$ ions in APH(3′)-IIIa are depicted as green spheres. The protein–ligand interactions in (**c, d**, and **e**) are depicted as yellow dotted lines.

Cph001 with a $K_D$ value $4.18 \pm 0.68 \mu M$, which is similar to the $K_D$ value for binding of CMN IIA (Supplementary Fig. 15).

**ATP and GTP binding and preferences in Cph001.** APHs and MPHs utilize ATP or GTP as phosphate donors for phosphorylation. Previous structural analysis have shown that ATP and GTP binding sites among these phosphotransferases are highly conserved, resulting in similar binding environments. For instance, in APH(2″)-IVa, the adenine base of ATP is stabilized through two hydrogen bonds: one between N6 and the main chain oxygen atom of Thr96 and another between N1 and the main chain nitrogen atom of Ile98 (Fig. 6a)[38]. However, the guanine base of GTP displays a slight positional shift relative to the adenine base of ATP. O6 of the guanine base, taking the place

of N1 of the adenine base, interacts with the main chain nitrogen atom of Ile98, while N1 of the guanine base interacts with the main chain oxygen atom of the same residue Ile98 (Fig. 6b)[38]. Notably, like APH(2″)-IVa, Cph001 can utilize both ATP and GTP as phosphate donors. Intriguingly, unlike typical APHs and MPHs, which reveal a hydrogen-bonding frameshift between ATP and GTP binding (Fig. 6a; Fig. 6b)[38], the crystal structures of Cph001$^{D189N}$-ATP and Cph001$^{D189N}$-GTP show identical binding positions for the adenine and guanine bases (Fig. 6c; Fig. 6d). Contrary to the hydrogen bonding of O6 of the guanine base with the main chain nitrogen atom in typical APHs and MPHs (Fig. 6b), N2 of the guanine base forms a hydrogen bond with the main chain oxygen atom of Val95 in Cph001 (Fig. 6d).

As mentioned earlier, enzyme kinetics reveal that Cph001 exhibits a preference for ATP as a phosphate donor. Each base of

**Table 1 Data collection and refinement statistics.**

| | Cph001[D189N] | Cph001[D189N]-CMN IIA | Cph001[D189N]-CMN IIB |
|---|---|---|---|
| Data collection | | | |
| Space group | P2₁2₁2₁ | P2₁2₁2₁ | P2₁2₁2₁ |
| Cell dimensions | | | |
| $a, b, c$ (Å) | 82.30, 86.94, 87.02 | 82.33, 84.49, 86.78 | 82.76, 85.15, 86.82 |
| $\alpha, \beta, \gamma$ (°) | 90.00, 90.00, 90.00 | 90.00, 90.00, 90.00 | 90.00, 90.00, 90.00 |
| Resolution (Å) | 30.00–2.30 (2.38–2.30) | 30.00–1.95 (2.02–1.95) | 30.00–2.20 (2.28–2.20) |
| $R_{sym}$ or $R_{merge}$ | 5.7 (58.2) | 5.6 (62.2) | 7.1 (44.5) |
| $I / \sigma I$ | 29.0 (3.4) | 35.7 (2.5) | 25.5 (2.9) |
| Completeness (%) | 99.9 (100.0) | 99.7 (97.4) | 99.0 (92.9) |
| Redundancy | 5.5 (5.6) | 8.0 (6.7) | 7.6 (6.2) |
| Refinement | | | |
| Resolution (Å) | 30.00–2.30 | 30.00–1.95 | 30.00–2.20 |
| No. reflections | 28419 | 44947 | 31648 |
| $R_{work}/R_{free}$ | 0.194/0.245 | 0.192/0.249 | 0.212/0.267 |
| No. atoms | | | |
| Protein | 4264 | 4229 | 4224 |
| CMN IIA | | 47 | |
| CMN IIB | | | 46 |
| Water | 188 | 376 | 201 |
| B-factors | | | |
| Protein | 37.2 | 24.2 | 38.4 |
| CMN IIA | | 28.9 | |
| CMN IIB | | | 38.4 |
| Water | 35.5 | 29.2 | 37.6 |
| R.m.s. deviations | | | |
| Bond lengths (Å) | 0.0044 | 0.0068 | 0.0035 |
| Bond angles (°) | 1.13 | 1.41 | 0.98 |

| | Cph001[D189N]-ATP | Cph001[D189N]-GTP | Cph001[D189N]-VIO |
|---|---|---|---|
| Data collection | | | |
| Space group | P2₁2₁2₁ | P2₁2₁2₁ | P2₁2₁2₁ |
| Cell dimensions | | | |
| $a, b, c$ (Å) | 82.83, 85.17, 87.34 | 82.14, 86.77, 87.32 | 82.76, 85.15, 86.82 |
| $\alpha, \beta, \gamma$ (°) | 90.00, 90.00, 90.00 | 90.00, 90.00, 90.00 | 90.00, 90.00, 90.00 |
| Resolution (Å) | 30.00–2.17 (2.26–2.17) | 30.00–2.30 (2.37–2.23) | 30.00–2.00 (2.07–2.00) |
| $R_{sym}$ or $R_{merge}$ | 4.5 (40.6) | 6.3 (43.7) | 5.3 (55.0) |
| $I / \sigma I$ | 41.1 (4.8) | 11.0 (2.4) | 36.5 (2.8) |
| Completeness (%) | 99.5 (95.5) | 99.5 (98.6) | 99.8 (99.3) |
| Redundancy | 7.3 (7.3) | 3.8 (3.7) | 7.8 (7.3) |
| Refinement | | | |
| Resolution (Å) | 30.00–2.17 | 30.00–2.23 | 30.00–2.00 |
| No. reflections | 33181 | 26881 | 43019 |
| $R_{work}/R_{free}$ | 0.196/0.255 | 0.231/0.288 | 0.205/0.252 |
| No. atoms | | | |
| Protein | 4226 | 4230 | 4245 |
| VIO | | | 48 |
| ATP | 31 | | |
| GTP | | 32 | |

**Table 1 (continued)**

| | Cph001[D189N]-ATP | Cph001[D189N]-GTP | Cph001[D189N]-VIO |
|---|---|---|---|
| Water | 247 | 36 | 142 |
| B-factors | | | |
| Protein | 36.8 | 64.9 | 47.9 |
| VIO | | | 59.9 |
| ATP | 38.1 | | |
| GTP | | 69.4 | |
| Water | 36.6 | 54.5 | 48.2 |
| R.m.s. deviations | | | |
| Bond lengths (Å) | 0.0030 | 0.0028 | 0.0031 |
| Bond angles (°) | 0.93 | 0.91 | 0.93 |

| | Cph001[D189N]-CMN IIA/ATP | Cph001[D189N]-VIO/ATP |
|---|---|---|
| Data collection | | |
| Space group | P2₁2₁2₁ | P2₁2₁2₁ |
| Cell dimensions | | |
| $a, b, c$ (Å) | 82.29, 87.04, 87.95 | 82.18, 87.39, 88.15 |
| $\alpha, \beta, \gamma$ (°) | 90.00, 90.00, 90.00 | 90.00, 90.00, 90.00 |
| Resolution (Å) | 30.00–3.00 (3.11–3.00) | 30.00–2.02 (2.14–2.02) |
| $R_{sym}$ or $R_{merge}$ | 14.5 (49.9) | 8.3 (58.4) |
| $I / \sigma I$ | 15.3 (4.5) | 10.4 (2.4) |
| Completeness (%) | 99.9 (100.0) | 99.4 (98.1) |
| Redundancy | 7.2 (7.4) | 3.9 (4.0) |
| Refinement | | |
| Resolution (Å) | 30.00–3.00 | 30.00–2.02 |
| No. reflections | 13117 | 37558 |
| $R_{work}/R_{free}$ | 0.212/0.275 | 0.218/0.252 |
| No. atoms | | |
| Protein | 4220 | 4230 |
| CMN IIA | 47 | |
| VIO | | 48 |
| ATP | 31 | 31 |
| Water | 27 | 155 |
| B-factors | | |
| Protein | 61.7 | 44.6 |
| Water | 32.9 | 40.3 |
| VIO | | 39.3 |
| CMN IIA | 71.9 | |
| ATP | 103.7 | 59.3 |
| R.m.s. deviations | | |
| Bond lengths (Å) | 0.0029 | 0.0075 |
| Bond angles (°) | 0.66 | 1.42 |

ATP and GTP bound in Cph001 respectively forms two hydrogen bonds with the protein backbone. For ATP, N6 interacts with the main chain oxygen atom of Ser93 and N1 interacts with the main chain nitrogen atom of Val95 (Fig. 6c). For GTP, N1 and N2 interact with the main chain nitrogen and oxygen atoms of Val95, respectively (Fig. 6d). There could be a variance in the strength of these hydrogen bonds when comparing adenine and guanine bases in their interactions with Cph001. In addition, the distance between O6 of the guanine base and the main chain oxygen atom of Ser93 is measured to be 3 Å (Fig. 6d). This proximity might induce a minor repulsion, potentially diminishing the binding affinity.

**Proposed catalytic mechanism of Cph001.** In every Cph001 structures, including apo and complex forms, the two polypeptide chains within the asymmetric unit adopt distinct conformations: an open form and a closed form (Fig. 7a). The two forms implicate conformational changes during enzyme catalysis. In all

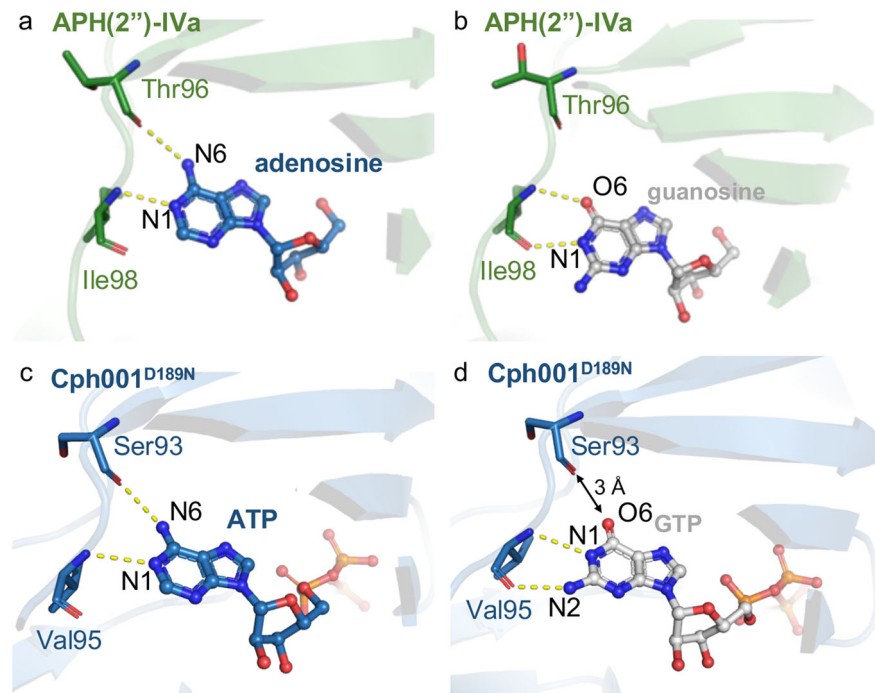

**Fig. 6 Local view of ATP- and GTP-binding sites in APH(2")-IVa[38] and Cph001. a** The crystal structure of the APH(2")-IVa-adenosine complex. **b** The crystal structure of the APH(2")-IVa-guanosine complex. **c** The crystal structure of Cph001[D189N]-ATP complex. **d** The crystal structure of Cph001[D189N]-GTP complex. ATP/adenosine and GTP/guanosine are depicted as balls and sticks in blue and gray. The protein–ligand interactions are depicted as yellow dotted lines.

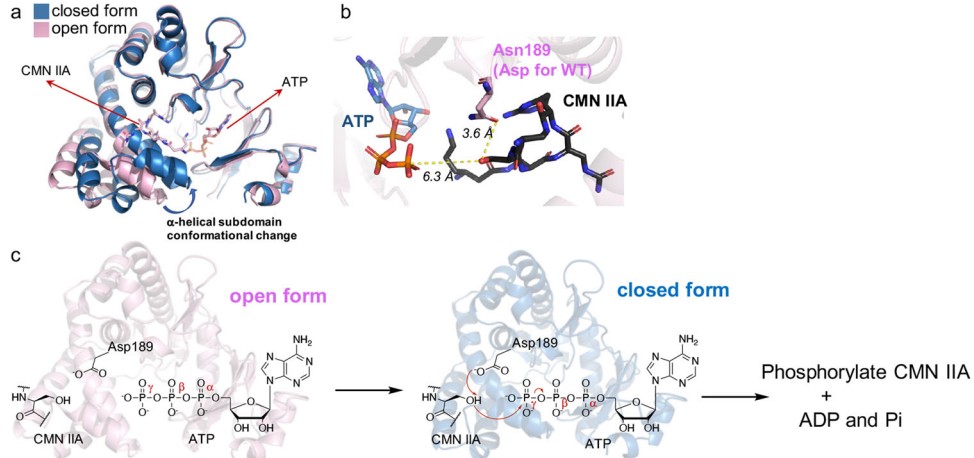

**Fig. 7 Proposed catalytic mechanism of Cph001. a** Superposition of the closed form and the open form of Cph001. The closed form and the open form are colored blue and pink, respectively. **b** Local view of the active site of Cph001. ATP and CMN IIA are depicted as sticks in blue and black. The interactions are depicted as yellow dotted lines. **c** Proposed catalytic mechanism of Cph001. Asp189 is ~3.6 Å distance from the hydroxyl group of CMN IIA, which points towards the γ-phosphate group of ATP. The open form of Cph001 accommodates CMN IIA and ATP, and catalysis occurs after a conformational change from the open form to the closed form. Asp189 acts as a general base to deprotonate the CMN IIA hydroxyl group, which subsequently attacks the γ-phosphate group of ATP for phosphorylation.

Cph001 structures, CMN IIA/VIO and ATP/GTP were only observed in the open form. In the open form of the ternary complex structure, Cph001[D189N]-CMN IIA/ATP, the hydroxyl group of CMN IIA is ~6.3 Å away from the γ-phosphate group of ATP (Fig. 7b). The closed form reveals that the α-helical sub-domain moves closer to the N-terminal lobe domain when compared to the open form (Fig. 7a). This suggests that a transition from the open to the closed form brings the two ligands together for catalysis. We propose that Cph001 undergoes a conformational change, wherein the α-helical subdomain

approaches the N-terminal lobe domain, allowing the substrate CMN IIA/VIO moves closer to ATP for the catalytic process. The carboxylate group of Asp189 (mutated to Asn in the current structure) forms a hydrogen bond with the hydroxyl group of CMN IIA/VIO at a distance of ~3.6 Å. Asp189 acts as a general base, deprotonating the hydroxyl group of CMN IIA/VIO to initiate the phosphorylation reaction (Fig. 7c).

**The putative CMN-resistance genes are found in a broad range of actinobacteria and distributed in human pathogens.** In this

study, 353 homologs of Cph were found and classified into five major clusters. SSN and bioinformatics analysis revealed that the 130 homologs in cluster I have highly conserved residues for CMN binding. Disk diffusion and enzyme activity assays supported that some proteins in this cluster confer resistance to CMN by antibiotic modification, specifically phosphorylation. Intriguingly, the Cph homologs in cluster I were discovered broadly in various bacterial species from 31 countries on five continents (Supplementary Fig. 17). Most notably, the putative CMN-resistance genes are present in various *Mycobacterium* and *Mycolicibacterium* species. These species belong to the non-tuberculosis *Mycobacterium* (NTM) group, which is known to induce lung disease and weaken the immune system[39]. Furthermore, the putative CMN-resistance genes have also been found in many human pathogens, including *Nocardia vulneris* (which causes disseminated nocardiosis)[40], *Mycolicibacterium fortuitum* (pulmonary infections)[41], *Mycobacterium septicum* (associated with catheter-related bacteraemia)[42], and *Actinomadura bangladeshensis* (*A. bangladeshensis*-induced foot mycetoma)[43]. These indicate that the putative CMN-resistance genes are distributed not only in the environmental bacteria but also human pathogens. Horizontal gene transfer from nonpathogens to pathogens, needless to say between the same genus, is known as one of the major pathways for ARGs transmission and dissemination. Our findings highlight an latent concern that the potential CMN-resistance genes may transfer to clinical TB pathogens, leading to drug resistance.

We further investigated whether other ARGs or mobile genetic elements exist nearby the three CMN-resistance genes, *cph001*, *cph002*, and *cph003*. For this purpose, we conducted an exhaustive ARGs detection analysis using AMRfinderPlus[44]. Notably, our results showed no presence of additional ARGs in close vicinity to our designated target genes, even within a 3000-nucleotide radius. Subsequently, we posited the possibility of mobile genetic elements such as transposons and insertion sequences, serving as mediators for the dissemination of these target genes across different species. However, a key finding from our study was the absence of any detectable long interspersed nuclear elements (LINEs) or short interspersed nuclear elements (SINEs) in the genomic areas our three target genes. No instances of any ARGs or mobile genetic elements were identified in the proximity to or within the genomic landscape of *cph001*, *cph002*, and *cph003*. This observation suggests that genetic mobility mechanisms may not be operative in the case of these three genes. Several factors might contribute to this observation: new mobile genetic elements that have not yet been identified, mobile genetic elements might be located further away from the resistance genes, or the resistance genes could be transferred through other mechanisms, such as generalized transduction. The lack of known mobile genetic elements near *cph001*, *cph002*, and *cph003* might be related to these resistance genes being predominantly found in actinomycetes species. Additional investigation is required to explore this intriguing question.

## Conclusion

Cph and Cph001 are peptide phosphotransferases (PPHs)[27]. APHs, MPHS, and PPHs share a similar structure and catalytic mechanism; however, they show different substrate-binding environment for substrate selectivity. The sequence-structure-function relationships that we evaluated suggest that proteins in cluster I specifically recognize and confer cross-resistance to CMN and its analog VIO. In contrast, proteins found in clusters II–V all lose the conserved residues for CMN binding, suggesting these proteins are unable to confer resistance to CMN. Putative residues involved in substrate binding are not conserved between

proteins in clusters II–V and APHs, MPHs, and PPHs. The substrate-binding environment suggests that proteins in clusters II–V represent phosphotransferase subfamilies with as yet unknown specificities. In summary, we identified and characterized ARGs specifically conferring resistance to CMN in nature. The bioinformatics, biochemical, and structural analysis supported that the proteins involved in cluster I are indeed related to CMN resistance. The CMN-resistance genes are found in a vast variety of bacteria in the world; notably, the putative CMN-resistance genes have been found in *Mycobacteria* species and several pathogens that are related to human diseases. Previous studies have reported that the wild bacterial species and the human microbiome may serve as a reservoir of antibiotic-resistance genes[1,45,46]. It is unclear how long these CMN-resistance genes have existed in such a wide array of bacteria. It also raises a question of why these unique resistance genes are in a broad range of actinobacteria even though the corresponding molecules are not in commercial or agriculture use[45]. Resistance genes may act as a microbial immune system, conferring resistance to toxic compounds from the environment. In this study, characterization of selected proteins from cluster I provides direct evidence for CMN resistance. In addition, this study also provides a further understanding of ARGs in nature and gives insights into combating the crisis of antibiotic resistance in the future.

## Methods

**Bioinformatics.** Over 3000 putative homologs of Cph, ranging from 15% to 100% amino acid sequence identities, were found using BLASTP analysis with the NCBI GenBank non-redundant protein sequence database. The collection of the Cph homologs for SSN analysis was achieved using the Enzyme Function Initiative (EFI) Enzyme Similarity Tool (EST)[47] with Cph as the query sequence for searching homologs with amino acid sequence identities among 20% to 70% from the UniProt database and was applied at an *e* value threshold of $10^{-55}$. The SSN was generated using EFI–EST and visualized in Cytoscape 3.9.1[48]. Consensus sequence logos among the Cph and its homologs were depicted using WebLogo 3[49].

**Gene cloning and site-directed mutagenesis of the Cph homologs.** The *cph001* (GenBank accession ID: WP_012891209), *cph002* (GenBank accession ID: MYT78782), *cph003* (GenBank accession ID: EWS93654), and *mphB* (GenBank accession ID: AB020531) genes were synthesized artificially and the *aph(3')-Ia* gene was amplified from pET28a. All genes were subcloned in expression vector pET22b (for disk diffusion assay) or pET28a (for protein production and purification). For the Cph001$^{D189A}$ and Cph001$^{D189N}$ variants, the plasmids were constructed by the QuikChange site-directed mutagenesis method. All plasmids used in this study were confirmed via DNA sequencing. The primers used for site-directed mutagenesis are listed in Table 2. Each of the wild-type and the mutant constructs was used to transform *E. coli* BL21 Rosetta (DE3), respectively, for gene expression and protein production.

**Gene expression, protein production, and purification of the Cph homologs.** A general procedure of gene expression is described as follows: 1 L of LB medium was inoculated with 10 mL of an overnight *E. coli* BL21 Rosetta (DE3) culture grown in LB medium containing 100 µg/mL ampicillin, induced with 1 mL 1 M isopropyl-β-ᴅ-1-thiogalactopyranoside (IPTG) at an OD$_{600}$ of 0.6. Cells were grown for a further 16 hr at 16 °C and were harvested by centrifugation at $6300 \times g$ for 30 min at 4 °C. The cells after gene expression and harvested by centrifugation were resuspended in lysis buffer (500 mM NaCl and 20 mM Tris,

| Table 2 Primers used for mutagenesis. | |
| --- | --- |
| **Primer Description** | **Sequence** |
| Cph001 D189A Forward | 5'- CGGTTGTCCATGGTGCTCTGGGTGCTGAAAATG -3' |
| Cph001 D189A Reversed | 5'- CATTTTCAGCACCCAGAGCACCATGGACAACCG -3' |
| Cph001 D189N Forward | 5'- GCGGTTGTCCATGGTAATCTGGGTGCTGAAAATG -3' |
| Cph001 D189N Reversed | 5'- CATTTTCAGCACCCAGATTACCATGGACAACCGC -3' |

pH 8.0). The cells were disrupted by sonication and were then centrifuged at $16,700 \times g$ for 30 min at 4 °C to remove cell debris. The Cph001 recombinant protein was purified using Ni-NTA affinity [HisTrap FF (Cytiva)] and size-exclusion chromatography [HiLoad Superdex 16/600 200 pg (Cytiva)] following standard procedures. All chromatography were performed by NGC Chromatography Systems (Bio-Rad). The protein purity was assessed by SDS-PAGE. All three variants of Chp001 recombinant proteins (Cph001, Cph001$^{D189N}$, and Cph001$^{D189A}$) were concentrated using Amicon Ultra-15 10,000 NMWL concentrators (Merck) in 400 mM NaCl and 20 mM Tris buffer at pH 8.0 for the following activity assay, enzyme kinetics, ITC analysis, and protein crystallization. Protein concentrations of Cph001, Cph001$^{D189A}$, and Cph001$^{D189N}$ were determined from the absorbance at 280 nm using the calculated molar absorption coefficient ($\varepsilon_{280} = 26,720\ M^{-1}\ cm^{-1}$). The pure Cph001 recombinant protein was stored at –80 °C.

**Disk diffusion assay**. The *E. coli* BL21 Rosetta (DE3) strains containing pET22b, and its carrying *cph*, *cph001*, *cph002*, *cph003*, *aph(3')-Ia*, and *mphB* plasmids were cultured overnight. The cells, respectively, were then diluted to 0.1 OD$_{600}$ in molten LB agar with 100 µg/mL ampicillin and 500 µM IPTG for the disk diffusion assay. The cells with ampicillin and IPTG were mixed well with molten LB agar and then poured into 10 cm diameter Petri dishes. The 7 mm disks were placed onto the LB agar and gently loaded with varying quantities of CMN IIA, CMN IIB, VIO, Kan, and Ery, ranging from 0 to 200 µg. The Petri dishes were placed in the incubator at 37 °C and the inhibition zones were observed after 24 hr.

**Size-exclusion chromatography analysis of Cph001**. Size-exclusion chromatography was carried out using a Superdex 200 10/300 GL column (Cytiva) with an NGC Chromatography Systems (Bio-Rad) at 4 °C. 1 mL of the pure Chp001 recombinant protein was loaded into the system in 400 mM NaCl and 20 mM Tris buffer at pH 8.0 at a flow rate of 0.45 mL/min. The column was calibrated with Gel Filtration Standard (Bio-Rad).

**Enzyme activity assay**. The phosphorylation reaction was measured by a coupled enzyme assay as following. The reaction 1, performed by 0.77 µM Cph001 with 2 mM antibiotic substrates (CMN IIA, CMN IIB, VIO, Kan, or Ery), was coupled to the reaction 2 and 3 in the presence of 4 U PK (pyruvate kinase from rabbit muscle, Merck), 4 U LDH (lactate dehydrogenase from rabbit muscle, Merck), 2 mM ATP or GTP, 2 mM PEP (phosphoenolpyruvate), 0.5 mM NADH, 40 mM KCl and 5 mM MgCl$_2$ in a reaction buffer containing 400 mM NaCl and 20 mM Tris buffer at pH 8.0. The reactions were mixed well in a final volume of 200 µL in 96-well plates. The Cph001 activity was assayed by determining the oxidation of NADH at 340 nm. The reaction buffer and enzyme concentration of the Cph reaction was followed by previous study[27]. The 340 nm absorbance changes were detected at 30 °C for every 1 min at a time in 15 min on an EPOCH Microplate Reader (BioTek). All reactions were carried out in triplicate.

**Kinetics of Cph001**. Enzyme kinetics of Cph001 were carried out under the same conditions of coupled enzyme assays using either a fixed concentration of ATP or GTP (2 mM) against variable concentrations of CMN IIA (0–6 mM), or using a fixed concentration of CMN IIA (20 mM) against variable concentrations of ATP (0 to 0.8 mM) or GTP (0 to 2 mM). The initial velocity data were plotted against substrate concentrations and fit the Michaelis-Menten equation. The $K_m$ and $V_{max}$ values were calculated based on the Michaelis-Menten equation. GraphPad Prism was used for statistical data analysis. All reactions were carried out in triplicate.

**Isothermal titration calorimetry (ITC) analysis of Cph001**. The dissociation constant of Cph001 against CMN IIA, CMN IIB, or VIO was determined using ITC (MicroCal iTC200, Malvern Panalytical). Each exothermic heat pulse was determined by an injection of 2 µL of 1 mM CMN IIA, CMN IIB, or VIO into 200 µL of 0.1 mM Cph001 protein solution in 400 mM NaCl and 20 mM Tris buffer at pH 8.0 at 25 °C. The integrated heat areas constitute a differential binding curve that was fitted with a standard single-site binding model by Origin 7.

**Protein crystallization and data collection**. Cph001$^{D189N}$ after protein purification was crystallized using the hanging drop vapor-diffusion method. Cph001$^{D189N}$ was concentrated to 12 mg/mL and was crystallized in 30% v/v glycerol, 5.6% w/v polyethylene glycol 4000, and 0.07 M sodium acetate trihydrate, pH 4.6, at 4 °C. All the complex structures, Cph001$^{D189N}$-CMN IIA, Cph001$^{D189N}$-CMN IIB, Cph001$^{D189N}$-ATP, Cph001$^{D189N}$-GTP, Cph001$^{D189N}$-VIO, Cph001$^{D189N}$-CMN IIA/ATP, Cph001$^{D189N}$-VIO/ATP, were obtained by soaking Cph001$^{D189N}$ crystals with 0.2 µL of 5 mM CMN IIA, CMN IIB, VIO, ATP, or GTP dissolved in the crystalized condition for overnight, respectively. All the diffraction data were collected at NSRRC on beamlines 15A, 05A, or 07A using a wavelength of 1 Å with MX300HE (05A and 07A beamlines) or MX300HS (15A beamline) detectors. Data were indexed and scaled using HKL2000 software[50].

**Structure determination and refinement**. The structure of Cph001$^{D189N}$ was solved by the molecular replacement (MR) method by MOLREP[51] using the structure of Cph (PDB entry 7F0A) as the search model[27]. The initial model was built using COOT[52] and refined with REFMAC[53]. The structure of Cph001$^{D189N}$ was used as the initial model for phasing the complex structures, including Cph001$^{D189N}$-CMN IIA, Cph001$^{D189N}$-CMN IIB, Cph001$^{D189N}$-ATP, Cph001$^{D189N}$-GTP, Cph001$^{D189N}$-VIO, Cph001$^{D189N}$-CMN IIA/ATP, and Cph001$^{D189N}$-VIO/ATP. The structural refinement of all complex structures was performed following the same steps of Cph001$^{D189N}$. The atomic coordinates and structural factors of all the structures have been verified and deposited in the Protein Data Bank (PDB) with the accession code 8I80 (Cph001$^{D189N}$), 8I82 (Cph001$^{D189N}$-CMN IIA), 8I84 (Cph001$^{D189N}$-CMN IIB), 8I85 (Cph001$^{D189N}$-ATP), 8I86 (Cph001$^{D189N}$-GTP), 8I89

(Cph001$^{D189N}$-VIO), 8I8G (Cph001$^{D189N}$-CMN IIA/ATP), and 8I8H (Cph001$^{D189N}$-VIO/ATP).

**Genomic context analysis.** Genomic sequences of the organisms containing the genes of interest (*cph001*, *cph002*, and *cph003*) were retrieved from the NCBI genome database. Antibiotic-resistance genes were identified within the genomic sequences using AMRfinderPlus[44], a dedicated tool adept at detecting a broad spectrum of ARGs. This analysis aimed to ascertain the presence of any ARGs in the proximity to the studied genes. Furthermore, to reveal potential mobile genetic elements such as transposons and insertion sequences in the regions encompassing the three target genes, RepearMasker[53] was utilized to identify LINEs and SINEs within the genomic sequences.

**Reporting summary.** Further information on research design is available in the Nature Portfolio Reporting Summary linked to this article.

## Data availability

The data associated with this study are available within the article, Supplementary Information and Supplementary Data 1. All the coordinates and structure factors have been deposited in the Protein Data Bank with accession codes 8I80, 8I82, 8I84, 8I85, 8I86, 8I89, 8I8G, and 8I8H. Source data are provided in this paper. All other data are available from the corresponding author on request.

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

## Acknowledgements

This work was supported by the Ministry of Science and Technology (MOST), Taiwan (grants 110-2113-M-A49-026-MY3 and 111-2113-M-002-025). This work is also supported by the Center for Intelligent Drug Systems and Smart Bio-devices (IDS²B) of NYCU, Taiwan. The authors thank the experimental facility and the technical services provided by the Synchrotron Radiation Protein Crystallography Facility of the National Core Facility Program for Biotechnology and the National Synchrotron Radiation Research Center (NSRRC), a national user facility supported by MOST, Taiwan. We thank Jeffrey Rudolf for the helpful discussions.

## Author contributions

C.Y.C. conceived the project, designed the experiments, and wrote the manuscript; S.I.T., W.T.L., P.Y.C., T.M.K., and C.Y.C. performed bioinformatics analysis; S.I.T., E.K.J., and Y.H.J. performed molecular cloning, protein production, protein purification, enzyme reactions, and disk diffusion assays; S.I.T., Y.C.P, Y.H.J., and Y.L.W. performed enzyme kinetics and ITC analysis; S.I.T., P.Y.H., and C.Y.C. collected the diffraction data and solved the crystal structures.

## Competing interests

The authors declare no competing interests.
