## [Peer Review File · Communications Biology]

Reviewers' comments:

Reviewer #1 (Remarks to the Author):

This study by Toh et al is focused on identification and functional characterization of new genes that confer resistance to the antibiotic capreomycin. This paper builds on previous work in the corresponding author's group where they performed biochemical and structural characterization of a capreomycin resistance enzyme, and they utilized the same suite of bioinformatic, biochemical and structural methodologies, this time focusing on homologous enzymes.

The importance of identification of new genes that confer antimicrobial resistance cannot be overstated, as we do not have nearly a complete understanding of the presence and dissemination of AMR genes in the bacterial kingdom. As such, I appreciated that the authors conducted this work as it adds to our knowledge of capreomycin resistance enzymes and provides novel insights into the enzyme family. I believe this paper will be of relevance and interest to the AMR community. I also commend the authors in their success in determining seven crystal structures – this provided essential data to understand the mechanism of their identified Cph enzyme homolog.

The experiments are largely performed soundly, and the conclusions are largely supported by the data, but I do have two major concerns that must be addressed since they may prohibit publication of the paper.

1. There are numerous issues with the disk diffusion assay. Most importantly, it is missing positive controls that would provide confidence this assay was done properly. For example, the strain expressing Cph enzyme itself could be shown for CMN IIA, CMN IIB and VIO plates, a strain expressing an Aph (aminoglycoside phosphotransferase) enzyme for the Kan plates, and a strain expressing a Mph (macrolide phosphotransferase) enzyme for the Mph plates. As well, the authors need to measure and report the sizes zone of inhibition around the antibiotic discs, rather than a "judgement call" by eye as it appears the authors did. It is hard to visualize the zone of clearing perhaps due to poor contrast in the images between the lawn and the zones of clearing - the authors should produce images with better contrast. A minor aspect about this assay is that the authors report micrograms of antibiotic in the discs, but it would be more appropriate to report micrograms per microliter.

2. I appreciated that the authors showed the Michaelis Menten kinetics curves, but the results show some concerns. The error bars are quite large for some points, and in 3 of the 4 curves there is at least one point that is significantly off the fitted saturation curve. As well, the fixed concentration of CMN IIA used for the ATP and GTP series is too low – it is only 6.8-fold and 4.3-fold the measured Km values of 0.8763 and 1.4030 mM – the accepted guideline is that the concentration of the ligand in excess should be >10-fold its measured Km values (<https://www.sciencedirect.com/science/article/pii/S2213020914000068?via=ihub>). I strongly encourage the authors to repeat this experiment again to address these concerns.

I have many other comments that are more minor but I also encourage the authors to address them:

-The authors should mention what was the source sequence database at the beginning of the Results and Discussion section and in the methods, as this was not included.

-Regarding SSN analysis: I thought there would be more family members than 353, this seems like a low number. How did the authors arrive at this number of sequences? The authors should also improve Figure 1 by showing how the families are related to each other in the SSN – as drawn, it

appears that the 5 clusters are totally distinct from each other – are there any links between the clusters? As well, in Figure 1, there are a lot of singletons – are these totally distinct family members without similarity to the other clusters? On Page 6, I would like the authors to mention the minimum sequence identity of the clusters II-IV to cluster I as it is important to know how diverse this sequence set is. Finally, I also recommend the authors show sequence alignments of each of the clusters, with Cph included at the top of the alignment and the CMN-binding residues labeled, as this would support their statement “Sequence analysis of proteins in cluster I revealed conservation of the CMN-binding residues (Fig. 2c); proteins in clusters II–IV do not conserve these residues.”

-Figure 1c – this would be improved by adding labels to the residues participating in CMN binding, and to ensure that this figure shows the catalytic residues.

-Regarding the disk diffusion experiment: why did the authors only select a small number of genes (3) for characterization? Are these representative of the sequence diversity in cluster 1, and if so the authors should state that, or was there another reason?

-It would be interesting for the authors to comment on the genomic context of the 3 genes that they studied, if available, as this would increase our awareness of the probability for mobilization of these genes across species. For example, are they in a region with other AMR genes? Are any on plasmids or do they have mobile genetic elements (transposons, insertion sequences) nearby?

-page 5: “Most proteins in this family were predicted to be phosphotransferases, including aminoglycoside phosphotransferases (APHs) and macrolide phosphotransferases (MPHs).” How was this prediction done? Are the authors referring instead to their (automated) annotations in the sequence database?

-page 8: what is Cph001 (inactive) in this figure? The authors should explain this in the main text.

-page 10: the authors mention that the Cph001 D189N variant as being less stable than the WT. The authors should explain what they mean by this, i.e. does the protein precipitate or degrade? And the authors should comment on whether this instability has an impact on the in vitro activity assay – i.e. does the protein degrade during the time to perform the activity assay? Finally, I recommend the authors change their wording for “mutant” when discussing a protein (mutant refers to DNA/RNA) and instead use the word “variant” when discussing the D189N protein.

-page 11: when discussing the similarity of the overall structures of Cph001 and Cph, the authors should mention the RMSD in angstrom over the number of C-alpha atoms.

-page 12: I found it quite interesting that ATP and GTP bind in the same position to Cph001-D189N, but the enzyme prefers ATP as the phosphate donor. This contrasts with the structurally-similar aminoglycoside phosphotransferases where these nucleotides have been shown to bind in different positions which explain the enzymes’ specificity ([https://www.jbc.org/article/S0021-9258\(20\)53050-0/fulltext](https://www.jbc.org/article/S0021-9258(20)53050-0/fulltext)). I encourage the authors to discuss/rationalize how the enzyme can be selective for ATP over GTP and whether there are subtle structural differences that could explain this.

-with respect to structure 8I86 Cph001-D189N with GTP: there is prominent negative Fo-Fc density at all 3 of the phosphates of GTP. Without having access to the crystal structure PDB and MTZ file, I cannot verify if these phosphates are actually present, what their B-factors are, or whether GTP was hydrolyzed. The authors should provide that PDB file for my review and should comment on this negative density.

-I am surprised that no crystal structure contains Mg²⁺ ions. The authors apparently did not include this in their purification or crystallization solutions. Did the authors attempt to soak Mg²⁺ into their crystals? Could they model it into their structures (based on other crystal structures) and comment on its binding interactions, which residues would interact with it?

-Figure 4: I recommend the authors label the site of phosphorylation on the antibiotics.

-page 14: The authors discuss distance between the carboxylate of Asp189 and hydroxyl group of the substrate as being 3 Å but they don't actually observe this as the crystal structure is of the Asn189 variant. The authors should reword to indicate that they assume the Asn189 variant is representative of the Asp189 WT sequence.

-page 15: The sentence "Kinase-mediated antibiotic resistance was carried out in aminoglycosides and macrolides antibiotics" does not make sense and should be reworded. Add Cph to the sentence "All APHs, MPHs, and Cph001 belong to the phosphotransferase family"

-page 15: I suggest adding the words "which is similar to the K_D value for binding of CPH IIA" to the end of the sentence "ITC analysis revealed that VIO binds to Cph001 with a K_D value 4.23 ± 0.67 μM (Supplementary Fig. 4)."

-I suggest the following section be moved to the end of the paper, as it is more of a "Discussion" type of section and would read better at the end: "Cph and Cph001 are peptide phosphotransferases (PPHs)²⁹. APHs, MPHs, and PPHs share a similar structure and catalytic mechanism; however, they show different substrate-binding environment for substrate selectivity. The sequence-structure-function relationships indicate that proteins in cluster I specifically recognize and confer cross-resistance to CMN and its analogue VIO. In contrast, proteins found in clusters II-IV all lose the conserved residues for CMN binding, suggesting these proteins are unable to confer resistance to CMN. Putative residues involved in substrate binding are not conserved between proteins in clusters II-IV and APHs, MPHs, and PPHs. The new substrate-binding environment implicated that proteins in clusters II-IV belong to the new phosphotransferase subfamilies."

-I also suggest some wording changes to that section:

Change "The sequence-structure-function relationships indicate that proteins in cluster I specifically recognize and confer cross-resistance to CMN and its analogue" to "The sequence-structure-function relationships that we evaluated suggest that proteins in cluster I specifically recognize and confer cross-resistance to CMN and its analogue"

Change "The new substrate-binding environment implicated that proteins in clusters II-IV belong to the new phosphotransferase subfamilies" to "The new substrate-binding environment suggests that proteins in clusters II-IV represent new phosphotransferase subfamilies with as yet unknown specificities."

-page 16: this sentence does not make sense to me and should be reworded: "known as a group of bacteria that causes rare lung infections and immune systems depression"

-page 16: I strongly suggest the authors reword the following sentence to be less strong and less effusive, as I believe the words "crisis" and "widespread" are not warranted. "Our finding provides an alternative hidden crisis that the widespread CMN-resistant genes may transfer to the clinical pathogens of TB resulting in drug resistance." (change finding to findings)

Typos/grammar suggestions:

-throughout: "self-resistant" should be changed to "self-resistance"

-page 4: "that may become major roles" is not clear, the authors should rephrase.

-page 5: "Cph, alone, endows" – remove both commas.

-page 5: "3,000 homologues" – homolog has a specific meaning and without experimental evidence, it is not clear if these sequences truly are Cph homologs. I suggest changing this to "3,000 putative homologues"

-page 6: The authors state that "the CMN-resistant genes appear to be widespread in nature". I do not believe the data support this statement, as the authors observe that only cluster 1 in Figure 1 conserve the CMN-binding site, and those sequences in cluster 1 are restricted to the Actinomycetota phylum. Unless the authors prove that cluster 1 contains sequences from bacteria outside of this phylum, or, they show that the other clusters contain enzymes that confer CMN resistance, I encourage the authors to rephrase this statement to be less effusive.

-page 8: "We then performed mutagenesis on the conserved general base Asp189 to generate..." – it would be more accurate to change this to "We then performed mutagenesis on the codons coding for the conserved general base Asp189 to generate..."

-page 8: "On the other hand, the apparent Km of ATP and GTP and kcat of Cph001 in the presence of CMN IIA were determined to be" – this would read better if this was rephrased to "On the other hand, the apparent Km and kcat of ATP and GTP for Cph001 in the presence of saturating CMN IIA were determined to be"

-page 8: "cph001 is capable of endowing E. coli the resistance to CMN IIB" – change "the" to "with"

-page 8: "This result also echoed the observation of the disk diffusion assay that Cph001 confers resistance to CMN by chemical modification and physical sequestration." – delete "by chemical modification and physical sequestration" as the disk diffusion assay itself does not provide this level of evidence.

-page 10: add the word "radiation" after "synchrotron".

-page 10: change "was determined by the molecular replacement with Cph (PDB entry 7F0A)29 as an initial searching model. Two polypeptide chains were found and built in an asymmetric unit (Fig. 4a) ." to "was determined by the molecular replacement (MR) method, with the structure of Cph (PDB entry 7F0A)29 as an initial search model. Two polypeptide chains were found and built in the asymmetric unit (Fig. 4a)."

-page 10: change "the complex structures" to "ligand complex structures"

-page 10: change "respectively, were determined by the molecular replacement using the structure of" "respectively, were determined by the MR method using the structure of"

-page 11: change "as a searching model" to "as the MR search model"

-page 11: change "Crystal structures of Cph001 revealing the molecular details for the binding sites of CMN" to "Crystal structures of Cph001 reveal molecular details for the binding sites of CMN"

-page 11: change "the pentapeptide backbones of CMN IIA and CMN IIB in Cph001 pose a similar conformation" to "the pentapeptide backbones of CMN IIA and CMN IIB bind to Cph001 in a similar conformation"

-page 12: change "Comparison with the ATP-binding environment of Cph001 and Cph" should be changed to: "In a comparison of the ATP-binding environments of Cph0001 and Cph"

- Figure 4: legend - change "Overall structure of Cph001 monomer" to "Overall structure of one Cph001 subunit" (apparently the protein is never monomeric so this inaccurate). 2 typos: "lob" domain should be "lobe"

-page 14: change "two polypeptide chains in an asymmetric unit" to "two polypeptide chains in the asymmetric unit". Change "subdomain moves closely to the N-terminal lobe domain with 7 Å transforming" to "subdomain move closer to" the N-terminal lobe domain by 7 Å as compared to"

-Figure 5: legend – Add the word "Proposed" before "Catalytic mechanism"

-page 15: section title "Cph001 conferring cross-resistance to CMN and VIO" should be changed to "Cph001 confers cross-resistance to CMN and VIO"

-page 15: change "E. coli harboring cph001 revealed resistance" to "E. coli harboring cph001 are resistant"

-page 15: change "We further performed couple enzyme assay to determine the phosphorylation" to "We further performed the couple enzyme assay to evaluate the phosphorylation"

-page 15: change "Moreover, an additional hydroxyl group at the" to "Moreover, the additional hydroxyl group at the"

-page 16: change "In this study, 353 homologues of Cph were found and classified into five major clusters. SSN and bioinformatics analysis revealed that 130 homologues in cluster I have very conserved residues for CMN binding" to "In this study, 353 homologues of Cph were found and classified into five major clusters. SSN and bioinformatics analysis revealed that the 130 homologues in cluster I have highly conserved residues for CMN binding"

-page 16: change "Disk diffusion and enzyme activity assays supported that the proteins" to "Disk diffusion and enzyme activity assays supported that some proteins"

-page 16: this occurs 3 times - change "Most notably, the putative CMN-resistant genes" to "Most notably, the putative CMN-resistance genes"

-page 16: change "disseminated nocardiosis" to "which causes disseminated nocardiosis"

-page 16: change "are distributed in not only the environmental bacteria" to "are distributed not only in the environmental bacteria"

-page 17: change "In summary, we fine-mapped and identified the ARGs" to "In summary, we identified and characterized ARGs"

Reviewer #3 (Remarks to the Author):

This work by Toh et al. uses sequence database search, sequence similarity analysis and clustering to identify putative capreomycin resistance genes (encoding homologues of a known capreomycin phosphotransferase) for subsequent biochemical and structural analysis. For the most part the studies are well done and are clearly reported (some issues are, however, noted below). While some new mechanistic insights are provided here, several elements do not represent a significant advance in understanding given the prior studies-with similar findings-of the enzyme (Cph) from the capreomycin producing bacterium. This is in part inherent in the nature of the results, but also in the writing, where I would encourage the authors to further emphasize the novelty of the new findings where they exist (rather than simply making comparisons to prior studies as is currently the case in many of the Results/ Discussion sub-sections).

Major comments.

1. Introduction (and mentioned elsewhere): capreomycin is no longer among recommended treatments for TB and e.g. no longer appears on the WHO's essential medicines lines. While the drug (and other related compounds) have a long history of use in this area, this discussion should be revised to reflect current practice.
2. The rationale for selecting Cph001, Cph002 and Cph003 from the 130 homologues in cluster 1 is not clear.
3. The Disk Diffusion assays do not appear to have been done according to standard protocol (with cells spread on the surface rather than embedded in the agar). While this does not negate comparisons within this work, the authors may consider noting this complication in comparison to other works. I also found it hard to see the zones of inhibition in the figure – impossible for some when printed, and not simple even with higher contrast on screen. When viewed on screen, the results for CMNIIB control and cph001 look the same to me (zones look a little “fuzzier” for the latter but are clearly still present and similar size to control). Finally, it is usual to measure the diameter of the zone of inhibition and classify (susceptible, intermediate, and resistant), adding this information would also be informative and aid in interpretation.
4. Many of the numerical data presented come with implausible degrees of accuracy (as implied by significant figures given for values and errors).
5. While I recognize the authors have previously reported the idea of resistance by sequestration for CMNIIB (which cannot be phosphorylated), the data do not fully support this mechanism as a biologically relevant form of resistance. In addition to the issue noted above with the DD assay, if growth in the presence of CMN is observed this is with high-level overexpression (induction with 0.5 mM IPTG). At a minimum, this is a significant caveat to this idea. With a K_d in the tens of micromolar range, it seems unlikely that neither protein expression level nor drug accumulation would result in a significant contribution to resistance via sequestration.
6. As a general comment (noted above), some additional description of the novelty of specific findings (the “Discussion” of Results and Discussion) would strengthen the manuscript in several places, with the end of section on p11/12 being a particularly good example – what did we learn here? (Presumably more than just that the previously characterized and current enzyme are essentially the same?) Similarly, the final section (“in summary”) is overly focused on the sequence analysis elements with little summarized about the advances in enzyme characterization.

7. The description on p14 of the open vs. closed conformations reflects one of the potentially most important advances here but left me confused. Several related comments/ questions:
 - a. What is the conformation of protein with only CMN/ VIO or ATP bound? Showing alignments of these structures in the supplemental might be helpful. Is the closed conformation exclusive to the structure with both ligands bound?
 - b. Why might only one of the two proteins in the asu have the closed conformation – is the movement of one limited by crystal packing contacts?
 - c. My interpretation from the descriptions provided (such as the long distance from CMN OH group and the phosphate) is that the authors were proposing that binding of both ligands triggers the conformational change necessary to bring them together for catalysis. However, it appears (line 5 and Fig. 5b?) that both ligands are only present in the open conformation? As such, what the authors mean by “undergoes a conformational change upon enzyme catalysis” needs to be more carefully explained.
8. I suggest that the final section on cross-resistance be supported by a separate figure that combines the relevant parts of Fig.3a,b and 4e (along with any duplicated data essential for comparison).
9. The ending of the cross-resistance section (top p16) seems out of place; this is about sequence conservation and would be better included in the text near Fig. 2.
10. Figure 6 is not necessary – could be moved to supplemental or simply removed (locations are provided in the supplemental table) and the current text at the Fig.6 callout more than adequately captures the idea that these enzymes are globally disseminated.
11. Systematic deviation from the fit late in some ITC titrations suggests that heats of dilution have not been removed from all data points before fitting. (This is best done using control titrations e.g. of ligand into buffer, but can also be estimated from the final points in the titration if binding has essentially reached saturation, which appears to be the case for most experiments here).

Minor comments (some have many instances throughout, where noted)

1. “antibiotic-resistant” should be “antibiotic-resistance” (first example in Abstract and many others).
2. Intro, para 1, line 4 – delete “It has been shown that”; this statement should instead/ in addition be supported by relevant citation(s).
3. Intro, para 1, line 7, “ARG” not “ARGs”
4. Figure 1 shows R on the chemical structure but lists R1 for the side chain definitions. Would also be good to mark the sites of modification and the named regions for Cph and Cac on these structures (Ser and beta-Lys)
5. P4 line 6 (and others), in “within the cmn biosynthetic gene cluster” cmn is the drug encoded and should be written Cmn (and not in italics)?
6. P7 section title, “confer” not “conferring”
7. Same section, line 8 – “used to transform” not “transformed into” (and elsewhere)
8. The colors/ symbols in Fig 3b are hard to distinguish (splitting the data as suggested above would help resolve this). It is also not clear to me what exactly “Cph001 (inactive)” actually is.
9. P10 line 6, reword “data were collected by synchrotron” (e.g. were collected at the X synchrotron on beamlines X,Y,Z (see Methods).”
10. P10, Line 8-9 (and elsewhere), “as the initial search model” not “an initial searching model”; also “the asymmetric unit” not “an asymmetric unit”.
11. P11, second section, line 11 “pose a similar conformation” is not clear to me – “are found in”?

12. Figure 4 – what is a “lob domain”? (image and legend); in legend “an” alpha-helical domain.
13. The first two sentences of the section beginning on p15 are awkward and unclear (what does resistance being carried out in the two drugs actually mean?).
14. P16 “immune system” not “immune systems”
15. P18 “plasmids used” not “plasmids sued” (and another “transformed with”)
16. P19 – give rcf (x g) for centrifugation steps (or rcf with rotor used). Centrifugation is used to remove cell debris (as a pellet) not “to remove the pellet”.
17. P19 – “assessed by SDS-PAGE” (not “justified”)?
18. P21 – “was subsequent for the” – missing word(s)?
19. Supp Fig3 – SD mentioned in legend but not given with values on plots (if they were, this would also highlight the issue with use of significant figures and the associated implied accuracy!)

Reviewers' comments:

Reviewer #1:

1. There are numerous issues with the disk diffusion assay. Most importantly, it is missing positive controls that would provide confidence this assay was done properly. For example, the strain expressing Cph enzyme itself could be shown for CMN IIA, CMN IIB and VIO plates, a strain expressing an Aph (aminoglycoside phosphotransferase) enzyme for the Kan plates, and a strain expressing a Mph (macrolide phosphotransferase) enzyme for the Mph plates. As well, the authors need to measure and report the sizes zone of inhibition around the antibiotic discs, rather than a "judgement call" by eye as it appears the authors did. It is hard to visualize the zone of clearing perhaps due to poor contrast in the images between the lawn and the zones of clearing - the authors should produce images with better contrast. A minor aspect about this assay is that the authors report micrograms of antibiotic in the discs, but it would be more appropriate to report micrograms per microliter.

We have repeated all the disk diffusion assays, adding all the positive controls suggested by the reviewer: *E. coli* expressing the Cph enzyme for CMN IIA, CMN IIB, and VIO plates; *E. coli* expressing the Aph(3')-Ia (aminoglycoside phosphotransferase) enzyme for the Kan plates; and *E. coli* expressing the MphB (macrolide phosphotransferase) enzyme for the Ery plates. Additionally, We captured enhanced contrast images of each disk and measured the diameter of the zone of inhibition (Fig. 3a and Fig. 4). To specify the quantity of antibiotics in the discs, we prefer to use micrograms, which directly indicate the amount in each disc.

2. I appreciated that the authors showed the Michaelis Menten kinetics curves, but the results show some concerns. The error bars are quite large for some points, and in 3 of the 4 curves there is at least one point that is significantly off the fitted saturation curve. As well, the fixed concentration of CMN IIA used for the ATP and GTP series is too low – it is only 6.8-fold and 4.3-fold the measured K_m values of 0.8763 and 1.4030 mM – the accepted guideline is that the concentration of the ligand in excess should be >10-fold its measured K_m values (<https://www.sciencedirect.com/science/article/pii/S2213020914000068?via=ihub>). I strongly encourage the authors to repeat this experiment again to address these concerns.

We thank the reviewer for the suggestions and corrections. We have repeated the enzyme kinetics according to the accepted guideline. In each experiment, the fixed concentration of the ligand exceeded its measured K_m value by more than 10 times. The K_m and k_{cat} values have been recalculated.

3. I have many other comments that are more minor but I also encourage the authors to address them:

3.1 -The authors should mention what was the source sequence database at the beginning of the Results and Discussion section and in the methods, as this was not included.

We searched for Cph homologues using the NCBI GenBank non-redundant protein sequence database and constructed an SSN using the UniProt database. The source sequence databases have been mentioned both at the beginning of the Results and Discussion section and within the methods on pg. 5 and pg. 21

3.2 -Regarding SSN analysis: I thought there would be more family members than 353, this seems like a low number. How did the authors arrive at this number of sequences? The authors should also improve Figure 1 (Fig. 2) by showing how the families are related to each other in the SSN – as drawn, it appears that the 5 clusters are totally distinct from each other – are there any links between the clusters? As well, in Figure 1 (Fig. 2), there are a lot of singletons – are these totally distinct family members without similarity to the other clusters? On Page 6, I would like the authors to mention the minimum sequence identity of the clusters II-IV to cluster I as it is important to know how diverse this sequence set is. Finally, I also recommend the authors show sequence alignments of each of the clusters, with Cph included at the top of the alignment and the CMN-binding residues labeled, as this would support their statement “Sequence analysis of proteins in cluster I revealed conservation of the CMN-binding residues (Fig. 2c); proteins in clusters II–IV do not conserve these residues.”

BLASTP analysis has identified over 3,000 Cph homologues with sequence identities ranging from 15% to 100%. We then collected the Cph homologues, with sequence identities between 20% and 70%, from the UniProt database to construct an SSN. The Bioinformatics part in the Methods section has been updated to elucidate our procedure for choosing Cph homologues for SSN analysis.

For Fig. 2, we included the sequence identities of Cph to cluster I–V. Cph shares the greatest sequence identity with proteins in cluster I, ranging from 36.9 to 68.9%. We also provide sequence alignments for each cluster, with Cph included at the top of the alignment (Supplementary Fig. 1–12). These sequence alignments reveal that each cluster exhibits distinct conserved residue patterns, differentiating them from one another. Furthermore, the sequence alignment of Cph with all the singleton proteins shows they have far fewer conserved residues than other clusters. This suggests that singleton proteins are markedly distinct from proteins in other clusters.

Additionally, we have labeled the CMN-binding residues in all sequence alignments (Supplementary Fig. 1–12) to validate the statement: “Sequence analysis of proteins in cluster I revealed conservation of the CMN-binding residues (Fig. 2c); proteins in clusters II–V do not conserve these residues”.

-Figure 1c (Fig. 2c) – this would be improved by adding labels to the residues participating in CMN binding, and to ensure that this figure shows the catalytic residues.

We have revised Fig. 2c, highlighting the residues involved in CMN binding and catalysis.

-Regarding the disk diffusion experiment: why did the authors only select a small number of genes (3) for characterization? Are these representative of the sequence diversity in cluster 1, and if so the authors should state that, or was there another reason?

Indeed, all proteins in cluster I exhibit a high conservation of the CMN-binding residues (Fig. 2c). We chose these three genes because their corresponding bacterial genomes have been sequenced, and none of them possess the CMN or VIO BGCs. Additionally, there are relevant reports on the genomes of these three bacterial strains (*Stand Genomic Sci* 2010, 2, 29; *Nat Commun* 2019, 10, 516; *Chembiochem* 2013, 14, 2345). There might be other genes in cluster I that have been reported, but in this study, we examined three of them and conducted a protein structure analysis on Cph001. We have added the relevant sentences

on pg. 7 to explain why we selected these genes for further experiments: “These three bacterial genomes were sequenced; notably, none of them possess the CMN or VIO BGCs.”.

-It would be interesting for the authors to comment on the genomic context of the 3 genes that they studied, if available, as this would increase our awareness of the probability for mobilization of these genes across species. For example, are they in a region with other AMR genes? Are any on plasmids or do they have mobile genetic elements (transposons, insertion sequences) nearby?

We appreciate the reviewer’s insightful comments and suggestions regarding our manuscript. We agree that investigating the genomic surroundings of these genes could provide valuable insights into their potential for mobilization across species and shed light on their roles in antibiotic resistance. Nevertheless, after conducting a thorough analysis, we did not identify any AMR genes or mobile genetic elements in close proximity to or within the genomic landscape of the three genes investigated in this study. This observation suggests that the mechanisms of genetic mobility might not be operative in the case of these three genes.

We have added our findings and comments into the Results and Discussion section (pg. 20) and have included a “Genomic context analysis” subsection in the Methods (pg. 25).

-page 5: “Most proteins in this family were predicted to be phosphotransferases, including aminoglycoside phosphotransferases (APHs) and macrolide phosphotransferases (MPHs).” How was this prediction done? Are the authors referring instead to their (automated) annotations in the sequence database?

The functions of most Cph homologues from NCBI database have not yet been confirmed. They are predicted and annotated in the sequence database.

We have updated the sentence on pg. 5 to: “Most putative Cph homologues identified through BLASTP analysis are annotated as phosphotransferases, including viomycin phosphotransferase, aminoglycoside phosphotransferase (APH), and macrolide phosphotransferase (MPH).”.

-page 8: what is Cph001 (inactive) in this figure? The authors should explain this in the main text.

We have included the annotation “(boiled enzyme)” following “inactive-Cph001” in the figure legend.

-page 10: the authors mention that the Cph001 D189N variant as being less stable than the WT. The authors should explain what they mean by this, i.e. does the protein precipitate or degrade? And the authors should comment on whether this instability has an impact on the in vitro activity assay – i.e. does the protein degrade during the time to perform the activity assay? Finally, I recommend the authors change their wording for “mutant” when discussing a protein (mutant refers to DNA/RNA) and instead use the word “variant” when discussing the D189N protein.

While we tried multiple conditions in our attempts to crystallize the Cph001 WT, all efforts were unsuccessful. Notably, we observed degradation bands in SDS-PAGE after storing the Cph001 WT at 4 °C for a week. This might be the reason for the inability to crystallize the

Cph001 WT. Nevertheless, this degradation did not impact our protein activity measurements, as we used freshly-prepared protein and conducted activity assays within 30 minutes. To provide clarity on the term “stable”, we have added an explanation on pg. 12 of the manuscript: “Degradation bands were observed on the SDS-PAGE for the wild-type Cph001 after a week of storage at 4 °C. This might be the reason why the wild-type Cph001 could not be crystallized.”

Regarding the term “mutant”, we have revised it based on reviewer’s suggestion and replaced it with “variant” when discussing the D189N protein.

-page 11: when discussing the similarity of the overall structures of Cph001 and Cph, the authors should mention the RMSD in angstrom over the number of C-alpha atoms.

We have incorporated the rmsd value into the text on pg. 13: “The overall structures of Cph001 and Cph revealed a similar fold with a root-mean-square deviations (rmsd) of 0.97 Å over a superposition of 215 C α atoms.”

-page 12: I found it quite interesting that ATP and GTP bind in the same position to Cph001-D189N, but the enzyme prefers ATP as the phosphate donor. This contrasts with the structurally-similar aminoglycoside phosphotransferases where these nucleotides have been shown to bind in different positions which explain the enzymes’ specificity ([https://www.jbc.org/article/S0021-9258\(20\)53050-0/fulltext](https://www.jbc.org/article/S0021-9258(20)53050-0/fulltext)). I encourage the authors to discuss/rationalize how the enzyme can be selective for ATP over GTP and whether there are subtle structural differences that could explain this.

We have added a section titled “ATP and GTP binding and preferences in Cph001” on pg. 16, discussing the selectivity of Cph001 for ATP and GTP. The binding modes of the guanine base of GTP in Cph001 and APH(2’)-IVa display notable differences. Based on the crystal structures of Cph001 in complex with ATP and GTP, we shed light on the reasons Cph001 has a preference for ATP over GTP as a phosphate donor.

-with respect to structure 8l86 Cph001-D189N with GTP: there is prominent negative Fo-Fc density at all 3 of the phosphates of GTP. Without having access to the crystal structure PDB and MTZ file, I cannot verify if these phosphates are actually present, what their B-factors are, or whether GTP was hydrolyzed. The authors should provide that PDB file for my review and should comment on this negative density.

We use phenix.refine to analyze the occupancy factor of GTP, which was found to be 0.79. Following this, we adjusted the occupancy factor of GTP and addressed the concern. The updated structure has been deposited on the PDB database, and the refinement statistics in Supplementary Table 2 have been revised. We also provide pdb and mtz files, along with validation reports ,for review.

-I am surprised that no crystal structure contains Mg²⁺ ions. The authors apparently did not include this in their purification or crystallization solutions. Did the authors attempt to soak Mg²⁺ into their crystals? Could they model it into their structures (based on other crystal structures) and comment on its binding interactions, which residues would interact with it?

We made multiple attempts to obtain the structure with Mg²⁺, employing methods like soaking and co-crystallization. However, these efforts were unsuccessful in obtaining the structure with Mg²⁺. Following the reviewer's suggestion, we examined the crystal structure of APH(3')-IIIa complexed with ADP and Mg²⁺. APH(3')-IIIa and Cph001 share a similar three-dimensional structure with an rmsd of 2.89 Å. We found that the Mg²⁺-binding residues, Asp208 and Asn195 in APH(3')-IIIa, correspond to Asp211 and Asn194 in Cph001, positioned proximal to the triphosphate group of ATP. Furthermore, these two residues are conserved across almost all the proteins in the SSN analysis (Supplementary Fig. 1–12), indicating their probable significance in Mg²⁺ binding. We have incorporated this insight into the manuscript on pg. 14 and Fig. 5g.

-Figure 4: I recommend the authors label the site of phosphorylation on the antibiotics.

Revised as suggested. The hydroxyl groups for phosphorylation have been labeled with arrows in Figure 5.

-page 14: The authors discuss distance between the carboxylate of Asp189 and hydroxyl group of the substrate as being 3 Å but they don't actually observe this as the crystal structure is of the Asn189 variant. The authors should reword to indicate that they assume the Asn189 variant is representative of the Asp189 WT sequence.

We have added a note, “(mutated to Asn in the current structure)”, after Asp189 to indicate that the Asn189 variant is representative of the Asp189 WT sequence.

-page 15: The sentence “Kinase-mediated antibiotic resistance was carried out in aminoglycosides and macrolides antibiotics” does not make sense and should be reworded. Add Cph to the sentence “All APHs, MPHs, and Cph001 belong to the phosphotransferase family”

This sentence has been revised to “Antibiotic resistance mediated by phosphotransferases was observed in many aminoglycoside and macrolide antibiotics” on pg. 10. “Cph” has been added to the sentence.

-page 15: I suggest adding the words “which is similar to the KD value for binding of CPH IIA” to the end of the sentence “ITC analysis revealed that VIO binds to Cph001 with a KD value 4.23 ± 0.67 μM (Supplementary Fig. 4).”

Revised as suggested.

-I suggest the following section be moved to the end of the paper, as it is more of a “Discussion” type of section and would read better at the end: “Cph and Cph001 are peptide phosphotransferases (PPHs)²⁹. APHs, MPHs, and PPHs share a similar structure and catalytic mechanism; however, they show different substrate-binding environment for substrate selectivity. The sequence-structure-function relationships indicate that proteins in cluster I specifically recognize and confer cross-resistance to CMN and its analogue VIO. In contrast, proteins found in clusters II–IV all lose the conserved residues for CMN binding, suggesting these proteins are unable to confer resistance to CMN. Putative residues involved in substrate binding are not conserved between proteins in clusters II–IV and APHs, MPHs,

and PPHs. The new substrate-binding environment implicated that proteins in clusters II–IV belong to the new phosphotransferase subfamilies.”

Revised as suggested. We have made the revision and relocated this paragraph to the conclusion section on pg. 20.

-I also suggest some wording changes to that section:

Change “The sequence-structure-function relationships indicate that proteins in cluster I specifically recognize and confer cross-resistance to CMN and its analogue” to “The sequence-structure-function relationships that we evaluated suggest that proteins in cluster I specifically recognize and confer cross-resistance to CMN and its analogue”

Revised as suggested.

Change “The new substrate-binding environment implicated that proteins in clusters II–IV belong to the new phosphotransferase subfamilies” to “The new substrate-binding environment suggests that proteins in clusters II–IV represent new phosphotransferase subfamilies with as yet unknown specificities.”

Revised as suggested.

-page 16: this sentence does not make sense to me and should be reworded: “known as a group of bacteria that causes rare lung infections and immune systems depression”

This sentence has been revised to “These species belong to the non-tuberculosis *Mycobacterium* (NTM) group, which is known to induce lung disease and weaken the immune system”.

-page 16: I strongly suggest the authors reword the following sentence to be less strong and less effusive, as I believe the words “crisis” and “widespread” are not warranted. “Our finding provides an alternative hidden crisis that the widespread CMN-resistant genes may transfer to the clinical pathogens of TB resulting in drug resistance.” (change finding to findings)

We have revised this sentence to “Our findings highlight an latent concern that the potential CMN-resistance genes may transfer to clinical TB pathogens, leading to drug resistance.”, The word “finding” has been updated to “findings”.

Typos/grammar suggestions:

-throughout: “self-resistant” should be changed to “self-resistance”

Revised as suggested.

-page 4: “that may become major roles” is not clear, the authors should rephrase.

This sentence has been revised to “Self-resistance determinants act as vital reservoirs and could significantly influence the evolution of pathogens toward drug resistance”.

-page 5: “Cph, alone, endows” – remove both commas.

Revised as suggested.

-page 5: “3,000 homologues” – homolog has a specific meaning and without experimental evidence, it is not clear if these sequences truly are Cph homologs. I suggest changing this to “3,000 putative homologues”

Revised as suggested.

-page 6: The authors state that “the CMN-resistant genes appear to be widespread in nature”. I do not believe the data support this statement, as the authors observe that only cluster 1 in Figure 1 conserve the CMN-binding site, and those sequences in cluster 1 are restricted to the Actinomycetota phylum. Unless the authors prove that cluster 1 contains sequences from bacteria outside of this phylum, or, they show that the other clusters contain enzymes that confer CMN resistance, I encourage the authors to rephrase this statement to be less effusive.

Revised as suggested. We have removed the word “widespread” and revised the sentence to “the Cph homologues in cluster I were found in a broad range of actinobacteria”.

-page 8: “We then performed mutagenesis on the conserved general base Asp189 to generate...” – it would be more accurate to change this to “We then performed mutagenesis on the codons coding for the conserved general base Asp189 to generate...”

Revised as suggested.

-page 8: “On the other hand, the apparent Km of ATP and GTP and kcat of Cph001 in the presence of CMN IIA were determined to be” – this would read better if this was rephrased to “On the other hand, the apparent Km and kcat of ATP and GTP for Cph001 in the presence of saturating CMN IIA were determined to be”

Revised as suggested.

-page 8: “cph001 is capable of endowing E. coli the resistance to CMN IIB” – change “the” to “with”

Revised as suggested.

-page 8: “This result also echoed the observation of the disk diffusion assay that Cph001 confers resistance to CMN by chemical modification and physical sequestration.” – delete “by chemical modification and physical sequestration” as the disk diffusion assay itself does not provide this level of evidence.

Revised as suggested.

-page 10: add the word “radiation” after “synchrotron”.

This sentence has been updated to “the diffraction data were collected at National Synchrotron Radiation Research Center (NSRRC) on beamlines 15A, 05A, or 07A.”

-page 10: change “was determined by the molecular replacement with Cph (PDB entry 7F0A)29 as an initial searching model. Two polypeptide chains were found and built in an asymmetric unit (Fig. 4a) .” to “was determined by the molecular replacement (MR) method, with the structure of Cph (PDB entry 7F0A)29 as an initial search model. Two polypeptide chains were found and built in the asymmetric unit (Fig. 4a).”

Revised as suggested.

-page 10: change “the complex structures” to “ligand complex structures”

Revised as suggested.

-page 10: change “respectively, were determined by the molecular replacement using the structure of” “respectively, were determined by the MR method using the structure of”

Revised as suggested.

-page 11: change “as a searching model” to “as the MR search model”

Revised as suggested.

-page 11: change “Crystal structures of Cph001 revealing the molecular details for the binding sites of CMN” to “Crystal structures of Cph001 reveal molecular details for the binding sites of CMN”

Revised as suggested.

-page 11: change “the pentapeptide backbones of CMN IIA and CMN IIB in Cph001 pose a similar conformation” to “the pentapeptide backbones of CMN IIA and CMN IIB bind to Cph001 in a similar conformation”

Revised as suggested.

-page 12: change “Comparison with the ATP-binding environment of Cph001 and Cph” should be changed to: “In a comparison of the ATP-binding environments of Cph0001 and Cph”

Revised as suggested.

- Figure 4: legend - change “Overall structure of Cph001 monomer” to “Overall structure of one Cph001 subunit” (apparently the protein is never monomeric so this inaccurate). 2 typos: “lob” domain should be “lobe”

Revised as suggested. We have also updated the text within the figure.

-page 14: change “two polypeptide chains in an asymmetric unit” to “two polypeptide chains in the asymmetric unit”. Change “subdomain moves closely to the N-terminal lobe domain with 7 Å transforming” to “subdomain move closer to” the N-terminal lobe domain by 7 Å as compared to”

Revised as suggested.

-Figure 5: legend – Add the word “Proposed” before “Catalytic mechanism”

Revised as suggested.

-page 15: section title “Cph001 conferring cross-resistance to CMN and VIO” should be changed to “Cph001 confers cross-resistance to CMN and VIO”

Revised as suggested.

-page 15: change “E. coli harboring cph001 revealed resistance” to “E. coli harboring cph001 are resistant”

Revised as suggested.

-page 15: change “We further performed couple enzyme assay to determine the phosphorylation” to “We further performed the couple enzyme assay to evaluate the phosphorylation”

Revised as suggested.

-page 15: change “Moreover, an additional hydroxyl group at the” to “Moreover, the additional hydroxyl group at the”

Revised as suggested.

-page 16: change “In this study, 353 homologues of Cph were found and classified into five major clusters. SSN and bioinformatics analysis revealed that 130 homologues in cluster I have very conserved residues for CMN binding” to “In this study, 353 homologues of Cph were found and classified into five major clusters. SSN and bioinformatics analysis revealed that the 130 homologues in cluster I have highly conserved residues for CMN binding”

Revised as suggested.

-page 16: change “Disk diffusion and enzyme activity assays supported that the proteins” to “Disk diffusion and enzyme activity assays supported that some proteins”

Revised as suggested.

-page 16: this occurs 3 times - change “Most notably, the putative CMN-resistant genes” to “Most notably, the putative CMN-resistance genes”

Revised as suggested.

-page 16: change “disseminated nocardiosis” to “which causes disseminated nocardiosis”

Revised as suggested.

-page 16: change “are distributed in not only the environmental bacteria” to “are distributed not only in the environmental bacteria”

Revised as suggested.

-page 17: change “In summary, we fine-mapped and identified the ARGs” to “In summary, we identified and characterized ARGs”

Revised as suggested.

Reviewer #3 (Remarks to the Author):

Major comments.

1. Introduction (and mentioned elsewhere): capreomycin is no longer among recommended treatments for TB and e.g. no longer appears on the WHO’s essential medicines lines. While the drug (and other related compounds) have a long history of use in this area, this discussion should be revised to reflect current practice.

We have updated the information and made revisions to the manuscript, adding the following paragraph to the introduction:

“CMN belongs to the tuberactinomycin family of antibiotics and has historically been a pivotal second-line drug for MDR-TB treatment. However, the growing concerns regarding treatment failures and relapses compared to regimens without it have diminished its preference. As a result, CMN is no longer recommended for MDR-TB treatment regimens. The World Health Organization (WHO) delisted CMN from the essential drugs roster in 2019. CMN now stands as a drug for conditional use.”

2. The rationale for selecting Cph001, Cph002 and Cph003 from the 130 homologues in cluster 1 is not clear.

Indeed, all proteins in cluster I exhibit a high conservation of the CMN-binding residues (Fig. 2c). We chose these three genes because their corresponding bacterial genomes have been sequenced, and none of them possess the CMN or VIO BGCs. Additionally, there are relevant reports on the genomes of these three bacterial strains (*Stand Genomic Sci* 2010, 2, 29; *Nat Commun* 2019, 10, 516; *Chembiochem* 2013, 14, 2345). There might be other genes in cluster I that have been reported, but in this study, we examined three of them and conducted a protein structure analysis on Cph001. We have added the relevant sentences on pg. 7 to explain why we selected these genes for further experiments: “These three bacterial genomes were sequenced; notably, none of them possess the CMN or VIO BGCs.”

3. The Disk Diffusion assays do not appear to have been done according to standard protocol (with cells spread on the surface rather than embedded in the agar). While this does not negate comparisons within this work, the authors may consider noting this complication in comparison to other works. I also found it hard to see the zones of inhibition in the figure – impossible for some when printed, and not simple even with higher contrast on screen. When viewed on screen, the results for CMNIIB control and cph001 look the same to me (zones look a little “fuzzier” for the latter but are clearly still present and similar size to control). Finally, it is usual to measure the diameter of the zone of inhibition and classify (susceptible, intermediate, and resistant), adding this information would also be informative and aid in interpretation.

We have repeated all the disk diffusion assays, adding all the positive controls: *E. coli* expressing the Cph enzyme for CMN IIA, CMN IIB, and VIO plates; *E. coli* expressing the Aph(3')-Ia (aminoglycoside phosphotransferase) enzyme for the Kan plates; and *E. coli* expressing the MphB (macrolide phosphotransferase) enzyme for the Ery plates. Additionally, We captured enhanced contrast images of each disk and measured the diameter of the zone of inhibition (Fig. 3a and Fig. 4). The regions delineating susceptible, intermediate, and resistant outcomes were not clearly defined in our experiments. Thus, we measured the affected rings to support whether the genes confer resistance to *E. coli*.

4. Many of the numerical data presented come with implausible degrees of accuracy (as implied by significant figures given for values and errors).

We have repeated the disc diffusion assay and captured enhanced contrast images of each disk and measured the diameter of the zone of inhibition (Fig. 3a and Fig. 4). Additionally, we conducted the enzyme kinetics again, adhering to the accepted guidelines. In each experiment, the fixed concentration of the ligand exceeded its measured K_m value by more than 10 times. The K_m and k_{cat} values have been recalculated.

5. While I recognize the authors have previously reported the idea of resistance by sequestration for CMNIIB (which cannot be phosphorylated), the data do not fully support this mechanism as a biologically relevant form of resistance. In addition to the issue noted above with the DD assay, if growth in the presence of CMN is observed this is with high-level overexpression (induction with 0.5 mM IPTG). At a minimum, this is a significant caveat to this idea. With a k_d in the tens of micromolar range, it seems unlikely that neither protein expression level nor drug accumulation would result in a significant contribution to resistance via sequestration.

In the Introduction, we mentioned that in addition to the phosphorylation mechanism employed by Cph, the CMN-producing strain features two other self-resistance strategies: *N*-acetylation facilitated by Cac (with the gene located outside the CMN BGC) and 16S rRNA methylation enabled by CmnU (with the gene located within the CMN BGC). Together, these three mechanisms afford the producing strain robust self-resistance to CMN through modifications to wither antibiotic or its target. The presence of multiple resistance genes likely ensures a consistent availability of resistance proteins to interact with or counteract CMN.

In our previous study, ITC analysis revealed that the K_D values for Cph interaction with CMN IIA and IIB are 16.11 μM and 5.48 μM , respectively (*ACS Chem. Biol.* 17, 138, 2022).

These values are comparable to or higher than the MIC of capreomycin, which ranges from 1–3 $\mu\text{g}/\text{mL}$ (~1.5–4.5 μM) (*Antimicrob. Agents Chemother.* 60, 4786, 2016 and *Antimicrob. Agents Chemother.* 62, e01724, 2018.). Even if only a small amount of the compound interact with Cph, the other two resistance proteins or alternative strategies might provide diverse pathways for coping with CMN.

In this manuscript, we focus on presenting data, including disc diffusion assay, K_D values, and the protein complex structure, to support that Cph001 can bind CMN IIB at high local concentration. We agree with the reviewer's caution against over-interpreting the data. We have removed the term "dual mechanism" and opted not to emphasize the sequestration mechanism.

6. As a general comment (noted above), some additional description of the novelty of specific findings (the "Discussion" of Results and Discussion) would strengthen the manuscript in several places, with the end of section on p11/12 being a particularly good example – what did we learn here? (Presumably more than just that the previously characterized and current enzyme are essentially the same?) Similarly, the final section ("in summary") is overly focused on the sequence analysis elements with little summarized about the advances in enzyme characterization.

Beyond addressing the resistance of the putative Cph homologues to CMN, we have expanded our discussion to provide a more in-depth analysis of sequence and SSN, as well as the sequence-structure-function relationships of Cph001. This includes examining the Mg^{2+} binding site, the preference for ATP and GTP binding, and the relationship between conformational changes and its catalytic mechanism.

For sequence and SSN analysis, we included the sequence identities of Cph to cluster I–V. We also provide sequence alignments for each cluster, with Cph included at the top of the alignment (Supplementary Fig. 1–12). These sequence alignments reveal that each cluster exhibits distinct conserved residue patterns, differentiating them from one another. Furthermore, the sequence alignment of Cph with all the singleton proteins shows they have far fewer conserved residues than other clusters. This suggests that singleton proteins are markedly distinct from proteins in other clusters.

To find the Mg^{2+} -binding site, we examined the crystal structure of APH(3')-IIIa complexed with ADP and Mg^{2+} . APH(3')-IIIa and Cph001 share a similar three-dimensional structure with an rmsd of 2.89 Å. We found that the Mg^{2+} -binding residues, Asp208 and Asn195 in APH(3')-IIIa, correspond to Asp211 and Asn194 in Cph001, positioned proximal to the triphosphate group of ATP. Furthermore, these two residues are conserved across almost all the proteins in the SSN analysis (Supplementary Fig. 1–12), indicating their probable significance in Mg^{2+} binding. We have incorporated this insight into the manuscript on pg. 14 and Fig. 5g.

We also investigate the genomic surroundings of these resistance genes. No instances of any AMR genes or mobile genetic elements in close proximity to or within the genomic landscape of *cph001*, *cph002*, and *cph003*. This observation suggests that the mechanisms of genetic mobility might not be operative in the case of these three genes. We have added our findings and comments into the Results and Discussion section (pg. 20)

Additionally, we have added a section titled "ATP and GTP binding and preferences in Cph001" on pg. 16, discussing the selectivity of Cph001 for ATP and GTP. The binding modes of the guanine base of GTP in Cph001 and APH(2'')-IVa display notable differences.

Based on the crystal structures of Cph001 in complex with ATP and GTP, we shed light on the reasons Cph001 has a preference for ATP over GTP as a phosphate donor.

7. The description on p14 of the open vs. closed conformations reflects one of the potentially most important advances here but left me confused. Several related comments/ questions:

- a. What is the conformation of protein with only CMN/ VIO or ATP bound? Showing alignments of these structures in the supplemental might be helpful. Is the closed conformation exclusive to the structure with both ligands bound?*
- b. Why might only one of the two proteins in the asu have the closed conformation – is the movement of one limited by crystal packing contacts?*
- c. My interpretation from the descriptions provided (such as the long distance from CMN OH group and the phosphate) is that the authors were proposing that binding of both ligands triggers the conformational change necessary to bring them together for catalysis. However, it appears (line 5 and Fig. 5b?) that both ligands are only present in the open conformation? As such, what the authors mean by “undergoes a conformational change upon enzyme catalysis” needs to be more carefully explained.*

We have re-examined the apo and complex structures of Cph001.

- In every Cph001 structures, including the apo and complex forms, the two polypeptide chains within the asymmetric unit exhibit different conformations: an open form and a closed form. Notably, CMN IIA/VIO and ATP/GTP were only observed in the open form. As suggested by the reviewer, we performed a structural superposition between the open and closed forms (Fig. 7a). The closed form revealed that the α -helical subdomain moves closer to the N-terminal lobe domain relative to its position in the open form.
- The crystal structure of the apo form Cph001 consists of both open and closed forms in the asymmetric unit. This suggests that the two conformations contribute to the crystal packing.
- The crystal structures of Cph001 revealed that CMN IIA/VIO and ATP/GTP were only observed in the open form. The closed form, however, blocks the entrance for the substrate and the phosphate donor (Fig. 7a). This suggests that Cph001 exhibits dynamic properties and can accommodate CMN IIA/VIO as well as ATP/GTP in its open form. However, since the conformations may be constrained within the crystal structure, neither CMN IIA/VIO nor ATP/GTP were observed in the closed form.
In the open form of the ternary structure, Cph001^{D189N-CMN IIA/ATP}, the hydroxyl group of CMN IIA is ~ 6.3 Å away from the γ -phosphate group of ATP (Fig. 7b). This indicates that a transition from the open to the closed form is essential to bring the two ligands in proximity for catalysis. Significantly, the closed form revealed that the α -helical subdomain responsible for CMN IIA binding moves closer to the N-terminal lobe domain relative to its position in the open form. This suggests a conformational change in Cph001, facilitating the substrate CMN IIA to approach ATP during the catalytic cycle (Fig. 7c).

We have revised Fig. 7 and the section “Proposed catalytic mechanism of Cph001” on pg. 17 to make the content clearer. The revised version is as follows:

“In every Cph001 structures, including apo and complex forms, the two polypeptide chains within the asymmetric unit adopt distinct conformations: an open form and a closed form. The closed form revealed that the α -helical subdomain moves closer to the N-terminal lobe domain when compared to the open form (Fig. 7a). Notably, CMN IIA/VIO and ATP/GTP were only observed in the open form. The closed form, however, blocks the entrance for the substrate and the phosphate donor. This suggests that Cph001 exhibits dynamic and can

accommodate CMN IIA/VIO as well as ATP/GTP in its open form. However, since the conformations may be constrained within the crystal structure, neither CMN IIA/VIO nor ATP/GTP were observed in the closed form.

In the open form of the ternary structure, Cph001^{D189N}-CMN IIA/ATP, the hydroxyl group of CMN IIA is ~6.3 Å away from the γ -phosphate group of ATP (Fig. 7b). This suggests that a transition from the open to the closed form is essential to bring the two ligands together for catalysis. We propose that Cph001 undergoes a conformational change, wherein the α -helical subdomain approaches the N-terminal lobe domain, allowing the substrate CMN IIA moves closer to ATP for the catalytic process. The carboxylate group of Asp189 (mutated to Asn in the current structure) forms a hydrogen bond with the hydroxyl group of CMN IIA/VIO at a distance of 3.6 Å (Fig. 7b). Asp189 acts as a general base, deprotonating the hydroxyl group of CMN IIA/VIO to initiate the phosphorylation reaction (Fig. 7c).

8. I suggest that the final section on cross-resistance be supported by a separate figure that combines the relevant parts of Fig.3a,b and 4e (along with any duplicated data essential for comparison).

We have reorganized the manuscript. The section discussing cross-resistance has been moved to the front, following the section “Activity assay and mutational analysis of Cph001 for CMN phosphorylation” on pg. 10.

Following the reviewer’s suggestions, we have separated Fig. 3 into two figures (Fig. 3 and Fig. 4). Furthermore, the disk diffusion assays in Fig. 3 and Fig. 4 have been repeated with adding all the positive controls: *E. coli* expressing the Cph enzyme for CMN IIA, CMN IIB, and VIO plates; *E. coli* expressing the Aph(3’)-Ia (aminoglycoside phosphotransferase) enzyme for the Kan plates; and *E. coli* expressing the MphB (macrolide phosphotransferase) enzyme for the Ery plates.

Regarding the crystal structure of Cph001 in complex with VIO, we prefer to consolidate this structure in Fig. 5 for structural comparison.

9. The ending of the cross-resistance section (top p16) seems out of place; this is about sequence conservation and would be better included in the text near Fig. 2.

Revised as suggested. The section discussing cross-resistance on pg.16 has been moved to the front, following the section “Activity assay and mutational analysis of Cph001 for CMN phosphorylation” on pg. 10.

10. Figure 6 is not necessary – could be moved to supplemental or simply removed (locations are provided in the supplemental table) and the current text at the Fig.6 callout more than adequately captures the idea that these enzymes are globally disseminated.

Revised as suggested. Fig. 6 has been moved to Supplementary Information as Supplementary Fig. 18.

11. Systematic deviation from the fit late in some ITC titrations suggests that heats of dilution have not be removed from all data points before fitting. (This is best done using control titrations e.g. of ligand into buffer, but can also be estimated from the final points in the titration if binding has essentially reached saturation, which appears to be the case for most

experiments here).

Revised as suggested. We use the point-by-point mode of subtracting reference (titration of CMN IIA, CMN IIB, or VIO into buffer) to process each of the ITC data. The heats of dilution have been removed from all data points before fitting. The K_D values and the resulting ITC graphs shown in Supplementary Fig. 16 have been updated.

Minor comments (some have many instances throughout, where noted)

1. *“antibiotic-resistant” should be “antibiotic-resistance” (first example in Abstract and many others).*

Revised as suggested.

2. *Intro, para 1, line 4 – delete “It has been shown that”; this statement should instead/ in addition be supported by relevant citation(s).*

Revised as suggested. We have deleted “It has been known that” and included the appropriate references, Ref. 3 and Ref. 4.

3. *Intro, para 1, line 7, “ARG” not “ARGs”*

Revised as suggested.

4. *Figure 1 shows R on the chemical structure but lists R1 for the side chain definitions. Would also be good to mark the sites of modification and the named regions for Cph and Cac on these structures (Ser and beta-Lys)*

We have revised the figure and legend. The residues Ser and β -Lys for Cph phosphorylation and Cac acetylation in CMN IIA and CMN IIB are colored red and green, respectively.

5. *P4 line 6 (and others), in “within the cmn biosynthetic gene cluster” cmn is the drug encoded and should be written Cmn (and not in italics)?*

Revised as suggested, change “*cmn*” to “CMN”.

6. *P7 section title, “confer” not “conferring”*

Revised as suggested.

7. *Same section, line 8 – “used to transform” not “transformed into” (and elsewhere)*

Revised as suggested.

8. *The colors/ symbols in Fig 3b are hard to distinguish (splitting the data as suggested above would help resolve this). It is also not clear to me what exactly “Cph001 (inactive)” actually is.*

We have regenerated Fig. 3b to enhance the differentiation among each dataset. Additionally, we have annotated “(boiled enzyme)” following “inactive-Cph001”.

9. P10 line 6, reword “data were collected by synchrotron” (e.g. were collected at the X synchrotron on beamlines X,Y,Z (see Methods).”

Revised as suggested. This sentence has been revised to “the diffraction data were collected at National Synchrotron Radiation Research Center (NSRRC) on beamlines 15A, 05A, or 07A.”

10. P10, Line 8-9 (and elsewhere), “as the initial search model” not “an initial searching model”; also “the asymmetric unit” not “an asymmetric unit”.

Revised as suggested.

11. P11, second section, line 11 “pose a similar conformation” is not clear to me – “are found in”?

We have revised this sentence to “the pentapeptide backbones of CMN IIA and CMN IIB are found in a similar conformation.”

12. Figure 4 – what is a “lob domain”? (image and legend); in legend “an” alpha-helical domain.

Revised as suggested. “lob” has been revised to “lobe” in image and legend and “a α -helical” has been revised to “an α -helical” in legend.

13. The first two sentences of the section beginning on p15 are awkward and unclear (what does resistance being carried out in the two drugs actually mean?).

This sentence has been revised to “Antibiotic resistance mediated by phosphotransferases was observed in many aminoglycoside and macrolide antibiotics” on pg. 10.

14. P16 “immune system” not “immune systems”

Revised as suggested.

15. P18 “plasmids used” not “plasmids sued” (and another “transformed with”)

Revised as suggested.

16. P19 – give rcf (x g) for centrifugation steps (or rcf with rotor used). Centrifugation is used to remove cell debris (as a pellet) not “to remove the pellet”.

Revised as suggested. 13,000 rpm and 8,000 rpm have been revised to 16,700 x g and 6,300 x g, and “the pellet” has been revised to “cell debris”.

17. P19 – “assessed by SDS-PAGE” (not “justified”)?

Revised as suggested.

18. P21 – “was subsequent for the” – missing word(s)?

We have revised this sentence to “The structure of Cph001^{D189N-apo} was used as the initial model for phasing the complex structures, including Cph001^{D189N-CMN IIA}, Cph001^{D189N-CMN IIB}, Cph001^{D189N-ATP}, Cph001^{D189N-GTP}, Cph001^{D189N-VIO}, Cph001^{D189N-CMN IIA/ATP}, and Cph001^{D189N-VIO/ATP}.”

19. Supp Fig3 – SD mentioned in legend but not given with values on plots (if they were, this would also highlight the issue with use of significant figures and the associated implied accuracy!)

We have repeated the enzyme kinetics according to the accepted guideline. In each experiment, the fixed concentration of the ligand exceeded its measured K_m value by more than 10 times. We have also added the standard error of the mean (SEM) values to the K_m and k_{cat} values.

Reviewers' comments:

Reviewer #1 (Remarks to the Author):

All my comments have been addressed satisfactorily.

Reviewer #2 (Remarks to the editor) should be in (Remarks to the Author):

"In the manuscript, Toh et.al. studied antibiotic resistant gene *cph* coded protein. The authors combined analysis of bioinformatics, biochemistry and crystal structures. This work provides important insights into the capreomycin resistant mechanisms of Cph protein.

I have only two questions regarding the crystallographic study section,

1, How the electron density surrounding the ligands (capreomycin viomycin) binding pocket look like?

2, In the supplementary table2, the authors described X-ray data collection. For Cph001D189N-CMN IIB crystal, the outer shell completeness is only 62.9% while the redundancy is 7.6. What is the reason for relatively low completeness in the outer shell ?"

Reviewer #3 (Remarks to the Author):

This manuscript by Toh et al. focuses on identification and molecular characterization of putative capreomycin resistance genes homologous to a known capreomycin phosphotransferase (*cph*). This is a revised manuscript in which the authors have made significant changes in response to the prior critiques. These changes have, for the most part, addressed my more significant concerns. A few issues remain, as noted below, which I would encourage the authors to consider.

1. R&D (p8) – resistance conferred by *cph0001/2/3* is described as similar to *cph* at different levels. This is confusing (see additional comment below), and the comparisons are hard to make, even with the improved image quality and bars indicating zones of clearance. The text showing the bar length is far too small and should be increased (drop the mm units, and mention in the legend, to give more room). However, the best way to simplify the comparisons would be to plot the data (e.g. bar graphs w/ average and individual data points).

2. My previous comment about use of excessive degrees of implied accuracy (sig. figures) was not completely addresses. On p9, values of K_m are given to 3 decimal places, which is clearly far beyond the accuracy of these data.

3. While the authors have generally done a good job of drawing out what is new in this study with the revisions made, this description of enzyme kinetics still needs some additional context – how do these values compare to Cph? What are the implications here. I also suggest that the plots used to derive these data be shown in a main figure (e.g. combined with current Fig. 2).

4. Two statements on p16/p17 are clearly contradictory: "In every Cph0001 structures,... the two polypeptide chains ... adopt distinct conformations: an open and a closed form" and "Notably, MNII/VIO and ATP/GTP were only observed in the open form". The discussion of conformations being constrained by crystal packing also does not really make sense because ALL ligands were soaked into

performed crystals of the apo protein! This does not necessarily change the argument that the open form is needed to accommodate both ligands, and may in fact add support to the idea that conformational changes are an important part of the mechanism (and can occur in the crystal). Transition to a closed(-like) conformation for catalysis WITH both ligands still remains a speculation, but I think it is a reasonable one based on all the evidence available.

5. In response to an excellent suggestion from the other reviewer to investigate the genetic environment of the Cph homologues, the authors conducted these studies. Perhaps somewhat surprisingly, no additional ARGs or evidence of other mobile genetic elements were identified. While this result is what it is, the descriptions make for an ending that is somewhat incongruous with the premise (set up) and overall messaging of this manuscript which is that cph is widely disseminated and part of a multi-drug resistance problem. The authors may wish to consider some additional discussion on the meaning/ implications of these new findings, and/ or to adjust other descriptions of cph's distribution and importance in antimicrobial resistance.

Minor/ text clarity issues:

1. Abstract: "We examined the function and targeted antibiotics on..."; I found this confusing, suggest rewording to: "We examined the function and antibiotics targeted by..."

2. Abstract: The following sentence has minimal information content: "We found that proteins in cluster I are from various species and confer resistance to CMN". Be more specific about which species (in line with other revisions)? CMN resistance is (at best) experimental validation that the SSN and clustering worked as expected – rephrase to say activity confirmed and focus sentence on the distribution of this gene family?

3. Introduction (p3, near end of second paragraph): "CMN belongs to the tuberactinomycin family...". This is true, but as written the statement might lead a naïve reader to think that VIO is not (prior descriptions to this point are about both CMN and VIO). Suggest, " Both drugs belong to... and CMN, in particular, has been a pivotal..."

4. Introduction (p4): define BGC at first use of the phrase (first sentence), and use BGC only where currently defined a few lines below.

5. R&D (p8): "similar to" ... unclear whether it is "resistance" itself or the "different levels" that are similar. Also, "confers resistance" – the strain(s) are the subject, so should read something like "becomes resistant when harboring" (the plasmid encoding a cph)

6. Fig 3 legend: Revise to "(b) Scheme showing the steps of the coupled enzyme assay used for Cph001 in this study."

7. R&D (p9): The second sentence of the section starting on page 9 is about the disk diffusion assays and is redundant with the sentence concluding the previous section. It can be deleted. Also suggest revising the description of attempts to characterize other enzymes, e.g. "To further confirm..., we attempted to produce Cph001, Cph002, and Cph003 in E. coli and to purify to homogeneity for enzyme assays. However, Cph002 and Cph003 were... Therefore, only Cph001 could be used..."

8. R&D (p12). New sentence "Degradation bands..." should be moved to one sentence earlier (before switching to the variant is mentioned)

9. R&D (p12). In naming the complexes, why are all ligands also superscripted? (It makes sense for

the aa substitution to be shown as superscript as this is an alteration within the protein; but, ligands should be shown as regular font). Suggest editing the sentence "Seven structures of Cph001D189N complexes were determined by MR using the apo protein as the search model: (list)." (Even without this edit, the word "respectively" should be deleted from this sentence.) Next sentence: "binary complex structures"... "were refined to resolutions between"

10. RMSD of <1Å over the whole protein is closer to "identical" than "similar"!

11. R&D (p14): revise "It has been demonstrated the significance..." to "The significance of Asp208 for... has been demonstrated". Below, "locate" should be "located"?

12. R&D (p16) "interacts with" (not "to"; at least two examples).

13. R&D (p20) "are existed" should be just "exist"?

Reviewer #1:

We thank reviewer 1 for their positive comments.

Reviewer #2:

We thank reviewer 2 for their positive comments and two suggestions. We address each of them as detailed below:

1. How the electron density surrounding the ligands (capreomycin viomycin) binding pocket look like?

The electron density surrounding the ligand pocket (CMN IIA, CMN IIB, VIO, ATP, and GTP) is clear and unambiguous. Please see the figures below. We have also provided pdb and mtz files of each complex structure for reference.

2. In the supplementary table 2, the authors described X-ray data collection. For Cph001D189N-CMN IIB crystal, the outer shell completeness is only 62.9% while the redundancy is 7.6. What is the reason for relatively low completeness in the outer shell?

In the supplementary Table 2, the outer shell completeness of Cph001^{D189N}-CMN IIB is 92.9%, not 62.9%.

Reviewer #3:

We thank reviewer 3 for carefully reading our manuscript and their suggestions, which have significantly enhanced the quality of this article. We address each of the requested suggestions as detailed below:

1. R&D(p8) – resistance conferred by cph001/2/3 is described as similar at different levels.

This is confusing (see additional comment below), and the comparisons are hard to make, even with the improved image quality and bars indicating zones of clearance. The text showing the bar length is far too small and should be increased (drop the mm units, and mention in the legend, to give more room). However, the best way to simplify the comparisons would be to plot the data (e.g. bar graphs w/ average and individual data points).

Revised as suggested. We deleted “similar to ...” and revised the sentence to “Disk diffusion assay revealed that each of the *E. coli* strains harboring *cph*, *cph001*, *cph002*, or *cph003*, becomes resistant to CMN IIA at different levels” on pg. 8. Additionally, we have updated Figure 3a and Figure 4 following the reviewer's suggestion by removing “mm” units and including this information in the legend to give more space.

Regarding Figure 3a, we prefer to display the disks to straightforwardly demonstrate that *E. coli* becomes less sensitive to CMN when it possesses the *cph001*, *cph002*, or *cph003* gene.

2. My previous comment about use of excessive degrees of implied accuracy (sig. figures) was not completely addresses. On p9, values of K_m are given 3 decimal places, which is clearly far beyond the accuracy of these data.

Revised as suggested. We have updated the K_m values in the text and figures to a reasonable two decimal places.

3. While the authors have generally done a good job of drawing out what is new in this study with the revisions made, this description of enzyme kinetics still needs some additional context – how do these values compare to Cph? What are the implications here. I also suggest that the plots used to derive these data be shown in a main figure (e.g. combined with current Fig. 2).

Revised as suggested. We have combined Supplementary Figure 15 (kinetics plots) with Figure 3. We also compare the K_m and k_{cat} values of Cph to that of Cph001. We have added additional discussion on pg. 10 as below:

“Previous kinetic studies have shown that the K_m value of Cph for CMN IIA with ATP as a phosphate donor ($K_m = 0.46$ mM) is on the same level as that of Cph001, being less than twice the K_m of Cph001. The environments of the CMN IIA binding sites in Cph and Cph001 are identical (the crystal structure of Cph001 will be discussed further below), leading to K_m values that are on the same level. However, the k_{cat} value of Cph for CMN IIA with ATP as a phosphate donor ($k_{cat} = 17.40$ min⁻¹) is about 27 times higher than that of Cph001. The more efficient catalysis of Cph phosphorylation stems from *cph* being the original self-resistance gene for CMN.”

4. Two statement on p16/p17 are clearly contradictory: “In every Cph001 structures,...the two polypeptide chains ... adopt distinct conformations: an open and a closed form” and “Notably, CMN/VIO and ATP/GTP were only observed in the open form”. The discussion of conformations being constrained by crystal packing also does not really make sense because ALL ligands were soaked into preformed crystals of the apo protein! This does not necessarily change the argument that the open form is needed to accommodate both ligands, and may in fact add support to the idea that conformational changes are an important part of the mechanism (and can occur in the crystal). Transition to a closed(-like) conformation for

catalysis WITH both ligands still remains a speculation, but I think it is a reasonable one based on all the evidence available.

Revised as suggested. We delete the sentences “The closed form, however, blocks the entrance for the substrate and the phosphate donor. This suggests that Cph001 exhibits dynamic properties and can accommodate CMN IIA/VIO as well as ATP/GTP in its open form. However, since the conformations may be constrained within the crystal structure, neither CMN IIA/VIO nor ATP/GTP were observed in the closed form”.

We have revised the section “Proposed catalytic mechanism of Cph001” on pg. 18 as follows: In every Cph001 structures, including apo and complex forms, the two polypeptide chains within the asymmetric unit adopt distinct conformations: an open form and a closed form (Fig. 7a). The two forms implicate conformational changes during enzyme catalysis. In all Cph001 structures, CMN IIA/VIO and ATP/GTP were only observed in the open form. In the open form of the ternary complex structure, Cph001^{D189N}-CMN IIA/ATP, the hydroxyl group of CMN IIA is ~6.3 Å away from the γ -phosphate group of ATP (Fig. 7b). The closed form reveals that the α -helical subdomain moves closer to the N-terminal lobe domain when compared to the open form (Fig. 7a). This suggests that a transition from the open to the closed form brings the two ligands together for catalysis. We propose that Cph001 undergoes a conformational change, wherein the α -helical subdomain approaches the N-terminal lobe domain, allowing the substrate CMN IIA/VIO moves closer to ATP for the catalytic process. The carboxylate group of Asp189 (mutated to Asn in the current structure) forms a hydrogen bond with the hydroxyl group of CMN IIA/VIO at a distance of ~3.6 Å. Asp189 acts as a general base, deprotonating the hydroxyl group of CMN IIA/VIO to initiate the phosphorylation reaction (Fig. 7c).

5. In response to an excellent suggestion from the other reviewer to investigate the genetic environment of the Cph homologues, the authors conducted these studies. Perhaps somewhat surprisingly, no additional ARGs or evidence of other mobile genetic elements were identified. While this result is what it is, the descriptions make for an ending that is somewhat incongruous with the premise (set up) and overall messaging of this manuscript which is that cph is widely disseminated and part of a multi-drug resistance problem. The authors may wish to consider some additional discussion on the meaning/ implications of these new findings, and/ or to adjust other descriptions of cph's distribution and importance in antimicrobial resistance.

In this study, the available data do not allow us to provide conclusions regarding the absence of mobile genetic elements in proximity to the resistance genes. Therefore, we do not want to speculate too much in this study. We have added some additional discussion on pg. 20 as below:

“Several factors might contribute to this observation: new mobile genetic elements that have not yet been identified, mobile genetic elements might be located further away from the resistance genes, or the resistance genes could be transferred through other mechanisms, such as generalized transduction. The lack of known mobile genetic elements near *cph001*, *cph002*, and *cph003* might be related to these resistance genes being predominantly found in actinomycetes species. Additional investigation is required to explore this intriguing question.”

Minor/ text clarity issues:

1. Abstract: “We examined the function and targeted antibiotics on...”; I found this confusing, suggest rewording to :”We examined the function and antibiotics targeted by...”

Revised as suggested.

2. Abstract: The following sentence has minimal information content: “We found that proteins in cluster I are from various species and confer resistance to CMN”. Be more specific about which species (in line with other revisions)? CMN resistance is (at best) experimental validation that the SSN and clustering worked as expected – rephrase to say activity confirmed and focus sentence on the distribution of this gene family?

Revised as suggested.

The sentence has been revised to “Sequence Similarity Network (SSN) analysis classified 353 Cph homologues into five major clusters, where the proteins in cluster I were found in a broad range of actinobacteria. We examined the function and antibiotics targeted by three putative resistance proteins in cluster I via biochemical and protein structural analysis. Our findings reveal that these three proteins in cluster I confer resistance to CMN, highlighting a significant aspect of CMN resistance within this gene family.”

3. Introduction (p3, near end of second paragraph): : CMN belongs to the tuberactinomycin family...”. This is true, but as written the statement might lead a naïve reader to think that VIO is not (prior descriptions to this point are about both CMN and VIO). Suggest, “Both drug belong to ... and CMN, in particular, has been a pivotal...”

Revised as suggested.

The sentence has been revised to “Both drugs belong to the tuberactinomycin family of antibiotics and CMN, in particular, has historically been a pivotal second-line drug for MDR-TB treatment.”

4. Introduction (p4): define BGC at first use of the phrase (first sentence), and use BGC only where currently defined a few lines below.

We have made revisions and defined BGC at first use of “biosynthetic gene cluster”.

5. R&D (p8): “similar to” ... unclear whether it is “resistance” itself or the “different levels” that are similar. Also, “confer resistance” – the strain(s) are the subject, so should read something like “becomes resistant when harboring” (the plasmid encoding a cph)

We delete “similar to ...” and revise “confers resistance” to “becomes resistant”.

The sentence has been revised to “Disk diffusion assay revealed that each of the *E. coli* strains harboring *cph*, *cph001*, *cph002*, or *cph003*, becomes resistant to CMN IIA at different levels (Fig. 3a)”

6. Fig 3 legend: Revise to “(b) Scheme showing the steps of the coupled enzyme assay used for Cph001 in this study.”

Revised as suggested.

7. R&D (p9): *The second sentence of the section starting on page 9 is about the disk diffusion assays and is redundant with the sentence concluding the previous section. It can be deleted. Also suggest revising the description of attempts to characterize other enzymes, e.g. “To further confirm..., we attempted to produce Cph001, Cph002, Cph003 in E. coli and to purify to homogeneity for enzyme assays. However, Cph002 and Cph003 were... Therefore, only Cph001 could be used...”*

Revised as suggested.

The repeated sentence has been deleted.

The sentences regarding the description of attempts to characterize other enzymes have been revised to “To further confirm their mechanism of resistance, we attempted to produce Cph001, Cph002, and Cph003 in *E. coli* and to purify to homogeneity for enzyme assays. However, Cph002 and Cph003 were not stable, undergoing aggregation and degradation during protein purification. Therefore, only Cph001 could be used for further biochemical characterization and structural determination” on pg. 9.

8. R&D (p12). *New sentence “Degradation bands...” should be moved to one sentence earlier (before switching to the variant is mentioned)*

Revised as suggested.

9. R&D (p12). *In naming the complexes, why are all ligand also superscripted? (It makes sense for the aa substitution to be shown as superscript as this is an alteration within the protein; but, ligands should be shown as regular font). Suggest editing the sentence “Seven structures of Cph001^{D189N} complexes were determined by MR using the apo protein as the search model: (list).” (Even without this edit, the word “respectively” should be deleted from this sentence.) Next sentence: “binary complex structures”...“were refined to resolutions between”*

Revised as suggested.

For naming the complexes, we have changed the names “Cph001^{D189N}-apo, Cph001^{D189N}-CMN IIA, Cph001^{D189N}-CMN IIB, Cph001^{D189N}-VIO, Cph001^{D189N}-ATP, Cph001^{D189N}-GTP, Cph001^{D189N}-CMN IIA/ATP, and Cph001^{D189N}-VIO/ATP” to “Cph001^{D189N}, Cph001^{D189N}-CMN IIA, Cph001^{D189N}-CMN IIB, Cph001^{D189N}-VIO, Cph001^{D189N}-ATP, Cph001^{D189N}-GTP, Cph001^{D189N}-CMN IIA/ATP, and Cph001^{D189N}-VIO/ATP” in the manuscript and the supplementary information.

10. *RMSD of <1Å over the whole protein is closer to “identical” than “similar”!*

Revised as suggested. “a similar...” has been revised to “an identical” on pg. 13.

11. R&D (p14): *revise “it has been demonstrated the significance...” to “The significance of Asp208 for... has been demonstrated”. Below, “locate” should be “located”?*

Revised as suggested.

12. R&D (p16) *“interacts with” (not “to”; at least two examples).*

Revised as suggested.

13. R&D (p20) *“are existed” should be just “exist”*

Revised as suggested.